SciPost Physics

Submission

# The Floquet Baxterisation

Yuan Miao[1,◇], Vladimir Gritsev[2,3], Denis V. Kurlov[3]

[1] Galileo Galilei Institute for Theoretical Physics, INFN,
Largo Enrico Fermi 2, 50125 Firenze, Italy
[2] Institute for Theoretical Physics, Universiteit van Amsterdam,
Science Park 904, Postbus 94485, 1090 GL Amsterdam, The Netherlands
[3] Russian Quantum Center, Skolkovo, Moscow 143025, Russia

◇yuan.miao@fi.infn.it

## Abstract

Quantum integrability has proven to be a useful tool to study quantum many-body systems out of equilibrium. In this paper we construct a generic framework for integrable quantum circuits through the procedure of Floquet Baxterisation. The integrability is guaranteed by establishing a connection between Floquet evolution operators and inhomogeneous transfer matrices obtained from the Yang–Baxter relations. This allows us to construct integrable Floquet evolution operators with arbitrary depths and various boundary conditions. Furthermore, we focus on the example related to the staggered 6-vertex model. In the scaling limit we establish a connection of this Floquet protocol with a non-rational conformal field theory. Employing the properties of the underlying affine Temperley–Lieb algebraic structure, we demonstrate the dynamical anti-unitary symmetry breaking in the easy-plane regime. We also give an overview of integrability-related quantum circuits, highlighting future research directions.

# 1 Introduction

Starting from the Onsager's exact solutions to the two-dimensional classical Ising model [1], exactly solvable models prove to be important and useful in many fields of theoretical physics, ranging from statistical mechanics [2,3] to high-energy physics [4]. Making use of techniques such as Bethe ansatz of quantum integrable models [3,5], one can obtain exact and sometimes mathematically rigorous results on physically relevant properties and quantities such as phase transitions and correlation functions. More recently, techniques in exactly solvable models have been used to study the out-of-equilibrium physics of quantum many-body systems, which is usually extremely difficult due to the astronomically large number of degrees of freedom involved. One particular example is the quantum quenches in integrable quantum spin chains that have been studied using the so-called quench action method [6,7], giving access to the late-time dynamics with an initial state away from any eigenstate. Another important achievement, the generalized hydrodynamics [8,9], allows us to study the transport properties of quantum integrable systems in an analytic and efficient manner.

Even though exactly solvable models have been successful to study the out-of-equilibrium physics of quantum many-body systems, applications to discrete space-time quantum systems are not yet fully developed. Specifically, quantum circuits become accessible with the recent advances on both theoretical and experimental sides [10], which is closely related to the stroboscopic evolution of Floquet systems. To begin with, a systematic understanding of the condition when quantum circuits can be solved using quantum integrability is still missing. In this work we present a general approach to this question by using the Floquet Baxterisation that shows explicitly the connection between a class of quantum circuits built with the R matrix from Yang–Baxter integrability and the inhomogeneous transfer matrices of integrable lattice statistical-mechanical models. This allows us to construct integrable (Floquet) circuits with arbitrary depth $n \in \mathbb{Z}_+$, cf. Fig. 3, using Yang–Baxter integrability. Our construction generalises the existing one for cases with depth $n = 2$. In addition, we exemplify the construction with a renowned example of the integrable Floquet circuit associated with the staggered 6-vertex model [2,11,12]. We use the Bethe ansatz technique to obtain the spectrum of the Floquet evolution operator. Using a connection with the underlying affine Temperley-Lieb algebraic structure we also show that in the easy-plane regime the system exhibits a dynamical phase transition associated with the breaking of a certain anti-unitary symmetry. Interestingly, the dynamical anti-unitary

symmetry breaking happens even with finite system sizes.

**Outline.** The structure of the paper is as follows. We start with a brief introduction to the crucial properties of the Yang–Baxter integrability in Sec. 2, which will be used extensively in the rest of the paper. Before getting to the main results, we present a brief overview of quantum circuits related to quantum integrability in Sec. 3. In Sec. 4 we present the method of Floquet Baxterisation. We rigorously derive the integrability condition for Floquet circuits and show the connection between the integrable Floquet circuits and the *inhomogeneous* transfer matrices of certain integrable lattice statistical-mechanical models with various boundary conditions. Then we move on to the example of the Floquet integrability related to the staggered 6-vertex model in Sec. 5. Using the Bethe ansatz technique, we present the spectrum of the integrable Floquet evolution operator in terms of the solutions to the Bethe equations. In Sec. 6 we then present a dynamical anti-unitary symmetry breaking for the integrable Floquet evolution operator that occurs in the easy-plane regime of the staggered 6-vertex model. Finally, in Sec. 7 we give an outlook and conclude.

## 2 The Yang–Baxter integrability

Before moving to the main results of the paper, in this section we briefly review the essential ingredients of the Yang-Baxter integrability.

We consider the R matrix $\mathbf{R}_{a,b}(u,v)$ acting on the Hilbert space $(\mathbb{C}^N)_a \otimes (\mathbb{C}^N)_b$ and satisfying the Yang–Baxter equation [2,3]

$$\mathbf{R}_{a,b}(u,v)\mathbf{R}_{a,c}(u,w)\mathbf{R}_{b,c}(v,w) = \mathbf{R}_{b,c}(v,w)\mathbf{R}_{a,c}(u,w)\mathbf{R}_{a,b}(u,v). \tag{2.1}$$

Hence we define the Lax operator $\mathbf{L}_{a,m}(u) = \mathbf{R}_{a,m}(u,u_m)$ with inhomogeneity $u_m \in \mathbb{C}$

$$\mathbf{R}_{a,b}(u,v)\mathbf{L}_{a,m}(u)\mathbf{L}_{b,m}(v) = \mathbf{L}_{b,m}(v)\mathbf{L}_{a,m}(u)\mathbf{R}_{a,b}(u,v), \tag{2.2}$$

which recovers the usual form of the Yang-Baxter relation for the Lax operator. Accordingly, we define another R matrix $\check{\mathbf{R}}_{a,b}(u,v)$ such that

$$\check{\mathbf{R}}_{a,b}(u,v) = \mathbf{R}_{a,b}(u,v)\mathbf{P}_{a,b}, \tag{2.3}$$

with the permutation operator $\mathbf{P}_{a,b}$ satisfying

$$\mathbf{P}_{a,b}\mathbf{F}_a\mathbf{P}_{a,b} = \mathbf{F}_b, \quad \mathbf{P}_{a,b}^2 = \mathbb{1}. \tag{2.4}$$

From Eq. (2.1), we then arrive at a different Yang-Baxter relation for $\check{\mathbf{R}}_{a,c}(u)$, which reads

$$\check{\mathbf{R}}_{a,b}(u,v)\check{\mathbf{R}}_{b,c}(u,w)\check{\mathbf{R}}_{a,b}(v,w) = \check{\mathbf{R}}_{b,c}(v,w)\check{\mathbf{R}}_{a,b}(u,w)\check{\mathbf{R}}_{b,c}(u,v). \tag{2.5}$$

The R matrix $\check{\mathbf{R}}_{a,b}(u,v)$ is the main building block for the integrable Floquet evolution operator.

Let us now define the inhomogeneous monodromy matrix with period $n \in \mathbb{Z}_+$ (with respect to lattice sites) and system size $L \bmod n = 0$ [2]

$$\mathbf{M}_a\big(u, \{u_j\}_{j=1}^n\big) = \prod_{m=1}^{L/n} \prod_{j=1}^n \mathbf{R}_{a,n(m-1)+j}(u,u_j), \quad u, u_1, u_2, \ldots u_n \in \mathbb{C}. \tag{2.6}$$

Then, the inhomogeneous transfer matrix with period $n$ acting on the physical Hilbert space can be defined as the partial trace of the inhomogeneous monodromy matrix over the auxiliary space,

$$\mathbf{T}\big(u, \{u_j\}_{j=1}^n\big) = \mathrm{Tr}_a\left[\mathbf{M}_a\big(u, \{u_j\}_{j=1}^n\big)\right]. \tag{2.7}$$

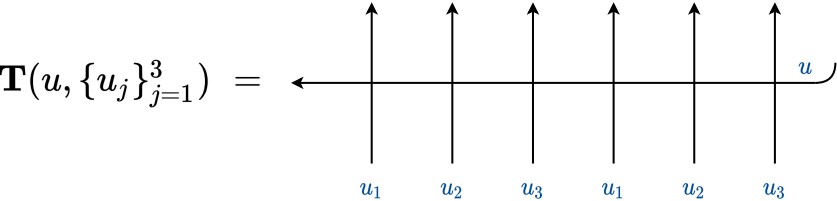

Figure 1: Diagrammatic demonstration of the R matrices and permutation operator.

$$\mathbf{T}\big(u, \{u_j\}_{j=1}^3\big) \;=\;$$

Figure 2: Diagrammatic demonstration of the inhomogeneous transfer matrix of period $n = 3$ with system size $L = 6$ and inhomogeneities $\{u_j\}_{j=1}^3$.

In the rest of the paper we mostly focus on the transfer matrices with periodic boundary condition. We shall make a few comments on the case of open boundary condition in Sec. 4.3.

From the Yang–Baxter equation (2.2), we have

$$\mathbf{R}_{a,b}(u,v)\mathbf{M}_a\big(u, \{u_j\}_{j=1}^n\big)\mathbf{M}_b\big(v, \{u_j\}_{j=1}^n\big) = \mathbf{M}_b\big(v, \{u_j\}_{j=1}^n\big)\mathbf{M}_a\big(u, \{u_j\}_{j=1}^n\big)\mathbf{R}_{a,b}(u,v), \quad (2.8)$$

which implies that the inhomogeneous transfer matrices are in involution, i.e.

$$\big[\mathbf{T}\big(u, \{u_j\}_{j=1}^n\big), \mathbf{T}\big(v, \{u_j\}_{j=1}^n\big)\big] = 0, \quad \forall u, v \in \mathbb{C}. \quad (2.9)$$

**Remark.** In order to obtain a *local* Floquet protocol that is *integrable*, we do not need to assume any property of the R matrix except that it satisfies the Yang-Baxter equation (2.1). In most of the previous works, see e.g. [13–20], it has been assumed that the R matrix also satisfies the regularity condition

$$\mathbf{R}_{a,b}(0,0) = \mathbf{P}_{a,b}, \quad (2.10)$$

and/or the difference form property,

$$\mathbf{R}_{a,b}(u,v) = \mathbf{R}_{a,b}(u - v), \quad (2.11)$$

both of which are not necessary for the integrability of the *local* Floquet protocol as proven below.

Note that one can represent the R matrix and the inhomogeneous transfer matrix in terms of graphs, as demonstrated in Figs. 1 and 2. This will be convenient for demonstrating some identities in Sec. 4.

## 3 Overview of quantum circuits related to concepts of integrability

Quantum circuits are well known tools in quantum information theory that are used to encode quantum computation with quantum gates, represented by a single, two- and/or

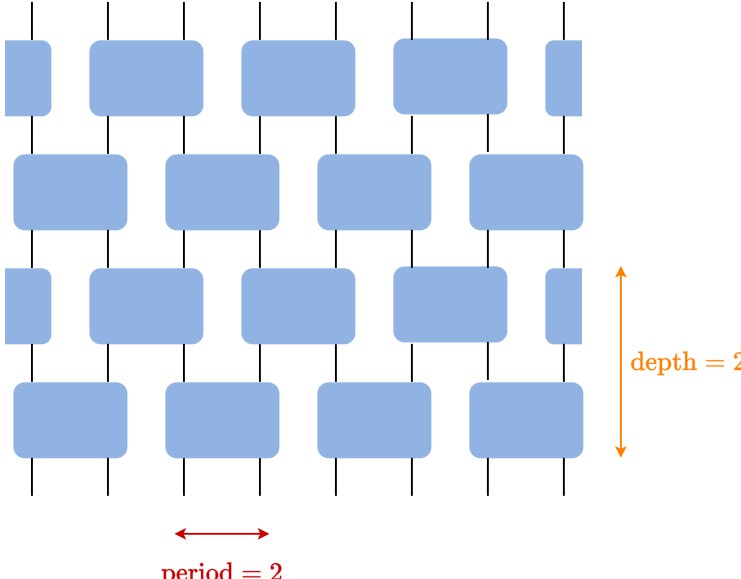

Figure 3: Demonstration of the brick-wall structure of the quantum circuit. Boxes correspond to two-qubit (qudit) gates and are realized by the Ř-matrices satisfying the Yang-Baxter equation. We denote the layer of quantum gates as the depth of the quantum circuit. For the brick-wall construction, the period of the quantum circuit with respect to the lattice sites equals the depth.

many-qubits or qudits, their initialization and measurements. Recently, there has been a surge of interest to certain aspects of quantum circuits related to many-body physics. One important class of circuits where dynamics can be computed explicitly is the case of random unitary circuits [21–24]. In this case quantum gates are represented by large unitary matrices typically taken from the circular unitary ensemble. Special attention has been paid to Floquet-type two-step protocols in which case one layer of the circuit is represented by a unitary $\mathbf{U}_1$ acting for a time $T_1$ between even-odd gates, followed by the second layer with $\mathbf{U}_2$ acting for a time $T_2$ between odd-even gates.

Here we focus on the opposite case of integrable quantum circuits. These integrable brick-wall circuits, see Fig. 3, are interesting for at least three reasons: (i) from the point of view of quantum information theory they serve as a viable tool for preparing some highly entangled states [25, 26]; (ii) they are suitable for benchmarking existing quantum computers and simulators, see [27–29] for recent examples; (iii) they could provide exact analytical tools for studying dynamical phases and phase transitions as demonstrated in this paper.

At the moment we feel that one should distinguish between at least three different cases of quantum circuits related to the concepts of quantum integrability. Below we provide a mixture of a brief overview of existing approaches, some new results and future perspectives.

## 3.1 Integrable Floquet protocols

This class of protocols is related to integrable lattice statistical-mechanical models. They represent a particular class of integrable Floquet dynamics introduced in [30]. These protocols rely on the analytic continuation of the Boltzmann weights of integrable stat-mechanical systems into the complex domain. In this case time steps $T_{1,2}$ are finite, and

except for the case of the Ising model are equal. The total unitary evolution operator is given by

$$\mathbf{U} \;=\; \prod_{i=1}^{N}(\mathbf{U}_{\mathrm{even}}(T/2)\mathbf{U}_{\mathrm{odd}}(T/2))_i \equiv \exp(-\mathrm{i}T\mathbf{H_F}) \tag{3.1}$$

$$\mathbf{U}_{\mathrm{even}} \;=\; \prod_{j=1}^{M}\check{\mathbf{R}}_{2j}(x,x_0), \qquad \mathbf{U}_{\mathrm{odd}} = \prod_{j=1}^{M}\check{\mathbf{R}}_{2j-1}(x,x_0), \tag{3.2}$$

where the parameter $x$ is some (known) function of time and $\mathbf{H_F}$ is the Floquet Hamiltonian. The central object here is the two-qubit (or qudit) gate, the $\check{\mathrm{R}}$ matrix that satisfies the celebrated Yang–Baxter equation, cf. (2.5). The proper definition of the Yang–Baxter integrability is explained in Sec. 2.

Many of the exactly solvable statistical-mechanical models are related to the so-called Temperley–Lieb (TL) algebra [2, 31]. It is defined by a set of two-site generators $e_{i,i+1} \equiv e_i$ that satisfy

$$e_i^2 = \beta e_i, \qquad e_i e_{i+1} e_i = e_i, \qquad [e_i, e_j] = 0, \quad \text{if} \quad |i - j| \geq 2. \tag{3.3}$$

In this case $x = \beta^{-1}(e^{\mathrm{i}T\beta/2} - 1)$ and $\check{\mathbf{R}}_j = 1 + xe_j$. Note that the generators of the TL algebra have numerous representations, including those in terms of the quantum Ising/Potts and the XXZ Hamiltonian densities.

Several first integrals of motion (i.e. a set of operators $Q_n$ that satisfy $[H_F, Q_n] = [Q_n, Q_m] = 0$) for the TL algebraic Floquet protocol have been found in Ref. [26], whose results hold regardless of the representation of the TL algebra. To our surprise, it appears that the first non-trivial integral

$$Q_1 = \sum_j e_j + a \sum_j (-1)^j[e_j, e_{j+1}] + b \sum_j \{e_j, e_{j+1}\}, \tag{3.4}$$

$$a = \frac{i}{2\beta}\sin(\beta T), \qquad b = -\frac{1}{\beta}\sin^2\left(\frac{\beta T}{2}\right), \tag{3.5}$$

when written in the representation of the XXZ Hamiltonian density, coincides exactly with the *Hamiltonian* of the lattice limit of the $SL(2,\mathbb{R})/U(1)$ black hole sigma-model. The latter, introduced in [32–34], describes a gauged version of the $SL(2,\mathbb{R})$ Wess–Zumino–Witten model and corresponds to a non-compact conformal field theory (CFT) with a continuum spectrum of scaling dimensions and a central charge $c = 2\frac{k+1}{k-1}$ related to the TL parameter $\beta$ as $k = \pi/\gamma$, where $\beta = -2\cos\gamma$. The model has been intensively studied in string theory literature, a highly incomplete list of papers includes [35–40].

We note that in the case of $\beta = 0$ the TL algebra has an infinite-dimensional $\mathfrak{sl}_2$ loop algebra structure of conserved charges and corresponds to a logarithmic CFT with the central charge $c = -2$ [41]. Both the lattice spin model and its relation to the continuum theory have been carefully studied by several groups, see [42–46]. The lattice spin model is related to an inhomogeneous 6-vertex model for a special choice of the inhomogeneity. Introduced by Baxter in [11, 12], this spin model and related ones have been intensively studied recently [47–53]. We also mention that inhomogeneous lattice models have been used by Destri and de Vega [13, 14] for defining the integrable lattice limit of relativistic field theories.

The connection between a non-compact/non-rational CFT, inhomogeneous lattice models and integrable Floquet quantum circuits is very promising and intriguing. It is fair to say that here we have an example of a *novel (and unusual) dynamical Floquet criticality*, since the Floquet Hamiltonian shares the same set of eigenstates as the operator $Q_1$ in Eq. (3.5), namely the spectrum of the CFT mentioned above.

### 3.2 Trotterized circuits

This type of circuits is constructed as follows:

$$\mathbf{U}(t) \quad = \quad \lim_{\delta t \to 0} \Big[ \prod_{i=1}^{N} (\mathbf{U}_{\text{even}}(\delta t) \mathbf{U}_{odd}(\delta t))_i \Big]^{t/\delta t}, \tag{3.6}$$

$$\mathbf{U}_{\text{even}} \quad \sim \quad \prod_{j=1}^{M} \check{\mathbf{R}}_{2j,2j+1}(\delta t), \qquad \mathbf{U}_{\text{odd}} \sim \prod_{j=1}^{M} \check{\mathbf{R}}_{2j-1,2j}(\delta t), \tag{3.7}$$

where $\delta t$ is an infinitesimal time step. For the most popular example of the integrable XXX spin chain [15] one has

$$\mathbf{U}_{\text{e/o}} = e^{-i\delta t \sum_{e/o} \text{h}_{\text{XXX}}} = \prod_{e/o} e^{iJ\delta t} \check{\mathbf{R}}_{e/o}(\tan 2J\delta t), \tag{3.8}$$

where the R-matrix acting between neighboring even-odd (e/o) sites is given by[1] $\check{\mathbf{R}}_{e/o}(\lambda) = (1 + i\lambda\mathbf{P})(1 + i\lambda)^{-1}$. This protocol has been used in several recent works: the multi-point correlation functions [17], and the temporal entanglement [54] have been computed; a dual unitary case of this circuit has been introduced and studied [19].

Strictly speaking, integrability in this type of circuits (guaranteed by the underlying integrable Hamiltonian) is achieved when the time step goes to zero. As pointed out in [27], the error in this circuit scales linearly with the Trotter time step i.e. with $\delta t$. While it can be cured by taking a smaller step size, this could result in an overall increase in the computation cost. Therefore, there is a need to balance accuracy and computation cost for which an efficient circuit compression is needed.

### 3.3 Protocols related to set-theoretic solutions of the Yang–Baxter equation

Interestingly, Yang–Baxter dynamics can be extended to a more generic and abstract setup. We remind the reader that the original quantum Yang–Baxter equation in (2.1) is defined for a linear operator $\mathbf{R}$ acting in the tensor product of two vector spaces $V \otimes V$: $\mathbf{R} : V \otimes V \to V \otimes V$.

In [55] Drinfeld suggested to consider a *set-theoretic* version of the Yang–Baxter equation defined as follows. Let $X$ be *any set* (perhaps endowed with a certain topology) and let $R : X \times X \to X \times X$ be a map from its square into itself[2]. Let $R_{ij} : X^n \to X^n$, with $X^n = X \times X \times \ldots \times X$, be the maps which act as $R$ on $i$th and $j$th factors and as an identity on the others. More precisely, if $R(x, y) = (f(x, y), g(x, y))$, where $x, y \in X$, then

$$R_{ij}(x_1, \ldots, x_n) = \Big( x_1, \ldots, x_{i-1}, f(x_i, x_j), x_{i+1}, \ldots, x_{j-1}, g(x_i, x_j), x_{j+1}, \ldots, x_n \Big) \tag{3.9}$$

for $i < j$ and

$$R_{ij}(x_1, \ldots, x_n) = \Big( x_1, \ldots, x_{i-1}, g(x_i, x_j), x_{i+1}, \ldots, x_{j-1}, f(x_i, x_j), x_{j+1}, \ldots, x_n \Big) \tag{3.10}$$

otherwise. In particular, for $n = 2$ one has $R_{21}(x, y) = (g(y, x), f(y, x))$. If $P : X^2 \to X^2$ is the permutation of $x$ and $y$: $P(x, y) = (y, x)$, then we obviously have $R_{21} = PRP$. In this setup, the set-theoretical Yang–Baxter equation reads

$$R_{12} \circ R_{13} \circ R_{23} = R_{23} \circ R_{13} \circ R_{12}. \tag{3.11}$$

---

[1]Operator $\mathbf{P}$ is the permutation operator, defined later in (4.10).

[2]To distinguish the set-theoretical R matrix from the quantum case we use here a different notation for the former.

The understanding of algebraic and geometric facets of the set-theoretic Yang–Baxter equation has been fairly well developed in mathematical literature [56–66], while its dynamical aspects have been considered in [67–69]. In addition, connection between the set-theoretic Yang–Baxter equation and integrable discrete-time dynamics has been studied in Refs. [70–72], and cellular automatons has been also anticipated, see [73] for a review. We believe that development of these ideas for quantum circuits is an interesting direction for future research, see [19, 74–76] for recent activity in this direction.

Let us give an example of a solution to the set-theoretical Yang–Baxter equation [3]. This is motivated by the generalised Fibonacci substitution rules $a \to b$, $b \to b^l a^k$, so that we consider the following ansatz for the functions $f(x, y)$ and $g(x, y)$ introduced above:

$$f(x, y) = x^n y^m, \qquad g(x, y) = x^p y^q. \tag{3.12}$$

The question is under what conditions of parameters $n, m, p, q$ this map is consistent with the set-theoretical Yang–Baxter equation (3.11).

Using the results of Appendix A, we come up with the following classes for arbitrary values of $n$ and $q$:

$$\begin{aligned}
A_{nq} &: (n, 0, 0, q), & B_f &: (n, 1 - nq, 0, q), \\
B_g &: (n, 0, 1 - nq, q), & P &: (0, 1, 1, 0).
\end{aligned} \tag{3.13}$$

Implementation of these classes in quantum dynamics will be considered elsewhere.

## 3.4 Remarks on the exponential solutions

Two-qudit quantum circuits that we are considering here correspond to the brick-wall structure shown in Fig. 3. An individual "brick" acts in the Hilbert space of two qudits, $V_j \otimes V_{j+1}$, and is supposed to satisfy the Yang–Baxter equation where spectral parameters are somehow related to time[4].

From the quantum dynamical perspective the individual gate $\check{\mathbf{R}}_j$ should represent the evolution operators on $V_j \otimes V_{j+1}$ and as such one could require that $\check{\mathbf{R}}_j$ would satisfy the semi-group property

$$\check{\mathbf{R}}_j(t_1)\check{\mathbf{R}}_j(t_2) = \check{\mathbf{R}}_j(t_1 + t_2), \quad \check{\mathbf{R}}_j(0) = 1, \tag{3.14}$$

for arbitrary time steps $t_{1,2}$. A natural ansatz for $\check{\mathbf{R}}_j(t)$ satisfying this requirement is an exponential solution

$$\check{\mathbf{R}}_{j,j+1}(t) = \exp(-itu_j), \tag{3.15}$$

for some operators $u_j$. In [77] (see also [78] for the relation of this construction to special types of symmetric polynomials) it was proven that exponential solutions of the Yang–Baxter equation satisfying the semi-group property (3.14) are exhausted by the operators defined by the following relations. Introducing $C_0(a, b) = a + b$ for adjacent operators $a = u_j$, $b = u_{j+1}$ and $C_i(a, b) = [a, C_{i-1}]$ for $i \geq 1$, it was shown that the minimal set of relations to fulfill the exponential solution of the Yang–Baxter equation is given by $C_j(a, b)$, which satisfies

$$[C_i(a, b), C_j(a, b)] = 0, \quad \forall i, j \geq 0. \tag{3.16}$$

There are many examples of algebraic structures satisfying the above relation. One of them is provided by the Hecke algebra [5]

$$\begin{aligned}
u_j u_{j+1} u_j &= u_{j+1} u_j u_{j+1}, & u_j^2 &= au_j + b, \\
u_j u_k &= u_k u_j, & |j - k| &> 1,
\end{aligned} \tag{3.17}$$

---

[3]To the best of our knowledge, this solution is new.

[4]In this subsection we focus on the Floquet protocol with depth $n = 2$.

[5]Note that the Temperley–Lieb algebra is a quotient of the Hecke algebra.

where one requires $b = 0$. Another interesting example is given by the universal enveloping algebra of upper triangular matrices with zero diagonal, $U_+(gl(n))$. In the latter case the generators satisfy the Serre relation $[u_i, [u_i, u_{i\pm1}]] = 0$. One more example is given by the Heisenberg algebra with $[u_i, u_{i+1}] = 0$. In relation to the Trotterized circuits, we note that if we require the semi-group relation to be satisfied only for infinitesimal times $t$ (i.e. for such $t_{1,2}$ that the products $t_1 t_2$ and higher orders can be neglected) then the exponential form can be generalized for $b \neq 0$ in Eq. (3.17). We note that in the present paper we do not require the regularity condition, e.g. the last identity in (3.14), which opens a room for more general R-matrices.

## 3.5 Some possible future directions

Up to now, we would like to point out a few directions that could be interesting to pursue further. In this respect we note that integrable quantum circuits can be understood in terms of the evolution/update of density matrices, not only as the unitary evolution of initial pure states.

- **Non-regular circuits and higher dimensions**. In his 1978 paper [79] R. Baxter introduced a solvable version of the eight-vertex model (which he called "Z-invariant model") on an *arbitrary lattice* formed by a planar set of intersecting straight lines provided that no three lines intersect at a point. In particular, this can be a Kagome lattice, detailed in [79], but any irregular lattice satisfying the property above works, quasicrystals in particular [80]. We also note that higher-dimensional generalizations are, at least in principle, possible using the Zamolodchikov tetrahedron equation [81]. We believe that these integrable cases are natural extensions of the regular brick-wall protocol in Fig. 3.

- **Monitored circuits.** Proliferation of entanglement, generated by the unitary time evolution, competes with entanglement collapse due to local projective measurements, which leads to the entanglement-type phase transitions. Various protocols were introduced recently in the realm of *random unitary circuits* [82–84]. While most of the works focus on numerics, we believe that monitored integrable circuits could provide some analytic insights into this type of transitions. Since measurement is defined by the insertion of *projection operators* (e.g. $(1 \pm \sigma^z)/2$ in the spin-1/2 case) at certain space-time points of a circuit, the problem may be translated into the task of evaluating multiple time-dependent *correlation functions*. For a spin-1/2 example above, a two-point projective measurement corresponds to a problem of evaluating the sum of zero-, one-, and two-point time-dependent correlation functions of $\sigma^z$ operators. While certain steady progress in computing these correlation functions for integrable spin lattice models exists, see e.g. [85–87] for recent developments, these results are still too complicated to extract any analytical insights.

- **Dissipative integrable circuits**. The concept of integrability can be extended to the realm of open quantum system dynamics of the density matrices [88–91]. Recently, several groups started investigating integrable dissipative circuits [18,20,92]. We believe that this is an interesting future direction by itself, which hopefully could unveil novel universality classes of dissipative quantum systems.

# 4 Floquet Baxterisation

In this section we present the procedure of the Floquet Baxterisation, i.e. finding the transfer matrices of certain integrable models that commute with the proposed Floquet evolution operators. We start with defining the Floquet evolution operator, describing the time evolution of a Floquet quantum circuit with depth $n \in \mathbb{Z}_+$. Then we prove that the previously defined Floquet evolution operator commutes with the inhomogeneous transfer matrices of a certain integrable vertex model, which contain one additional spectral parameter. Even though the procedure differs from the original Baxterisation construction by Jones [93], the philosophy here is the same. Namely, one introduces an additional spectral parameter to certain transfer matrices (in our case, the Floquet evolution operators), which are in involution guaranteed by the Yang–Baxter integrability.

Before we start to state the main theorems, we would like to make a few remarks. We note that previous works on the topic either focus on explicit examples [13–18] or make certain assumptions such as the regularity of the R-matrix [19,20], cf. (2.10). In this work, we present the most general construction of the Floquet Baxterisation without making the assumptions that the R-matrix satisfying the Yang–Baxter equation is regular or is of the difference form. Moreover, we generalise previous constructions that focus on the Floquet evolution operators with depth $n = 2$. Our results are more generic, extending to an arbitrary depth $n \in \mathbb{Z}_+$.

## 4.1 Floquet Baxterisation with periodic boundary condition: generic case

Equipped with the notion of Yang–Baxter integrability, we are ready to proceed by applying the Floquet Baxterisation to a Floquet evolution operator with depth $n \in \mathbb{Z}_+$ with periodic boundary condition.

**Theorem 1.** *The periodic Floquet evolution operator with depth $n$ and the system size $L$ satisfying $L \bmod n = 0$*

$$
\mathbf{U}_{\mathrm{F}}\big(\{u_j\}_{j=1}^n\big) = \prod_{k=n}^{1} \mathbf{V}_k\big(\{u_j\}_{j=1}^n\big),
$$

$$
\mathbf{V}_k\big(\{u_j\}_{j=1}^n\big) = \prod_{m=1}^{L/n} \prod_{j=1}^{n-1} \check{\mathbf{R}}_{n(m-1)+k+j,\,n(m-1)+k+j+1}(u_1, u_{j+1}).
$$

$$(4.1)$$

*is integrable, i.e.*

$$
\big[\mathbf{U}_{\mathrm{F}}\big(\{u_j\}_{j=1}^n\big), \mathbf{T}\big(u, \{u_j\}_{j=1}^n\big)\big] = 0, \qquad \forall u \in \mathbb{C},
$$

$$(4.2)$$

*where the inhomogeneous transfer matrix is defined in Eq. (2.7). The inhomogeneous transfer matrix $\mathbf{T}(u, \{u_j\}_{j=1}^n)$ is regarded as the baxterised Floquet evolution operator.*

We call Theorem 1 the Floquet Baxterisation, making the connection between the Floquet evolution operator (which can be considered as a tilted transfer matrix of the underlying integrable vertex model [94]) and the inhomogeneous transfer matrix with one additional spectral parameter.

Before we present the proof of Theorem 1, we would like to provide two explicit examples of the periodic Floquet evolution operators, in order to gain some intuition. To begin with, we consider the case with the depth $n = 2$ (system size $L \bmod 2 = 0$). The Floquet

evolution operator of depth $n = 2$ is $\mathbf{U}_{\mathrm{F}}\big(\{u_j\}_{j=1}^2\big) = \prod_{k=2}^1 \mathbf{V}_k\big(\{u_j\}_{j=1}^2\big)$, where $\mathbf{V}_k$ can be expressed as

$$
\begin{aligned}
\mathbf{V}_2\big(\{u_j\}_{j=1}^2\big) &= \check{\mathbf{R}}_{1,2}(u_1, u_2)\check{\mathbf{R}}_{3,4}(u_1, u_2)\cdots\check{\mathbf{R}}_{L-1,L}(u_1, u_2), \\
\mathbf{V}_1\big(\{u_j\}_{j=1}^2\big) &= \check{\mathbf{R}}_{L,1}(u_1, u_2)\check{\mathbf{R}}_{2,3}(u_1, u_2)\cdots\check{\mathbf{R}}_{L-2,L-1}(u_1, u_2).
\end{aligned}
\tag{4.3}
$$

Similarly, for the depth $n = 3$ (system size $L \bmod 3 = 0$), the Floquet evolution operator with depth $n = 3$ is $\mathbf{U}_{\mathrm{F}}\big(\{u_j\}_{j=1}^3\big) = \prod_{k=3}^1 \mathbf{V}_k\big(\{u_j\}_{j=1}^3\big)$, which consists of three parts,

$$
\begin{aligned}
\mathbf{V}_3\big(\{u_j\}_{j=1}^3\big) =& \big[\check{\mathbf{R}}_{1,2}(u_1, u_2)\check{\mathbf{R}}_{2,3}(u_1, u_3)\big]\big[\check{\mathbf{R}}_{4,5}(u_1, u_2)\check{\mathbf{R}}_{5,6}(u_1, u_3)\big]\cdots \\
& \times \cdots \big[\check{\mathbf{R}}_{L-2,L-1}(u_1, u_2)\check{\mathbf{R}}_{L-1,L}(u_1, u_3)\big], \\
\mathbf{V}_2\big(\{u_j\}_{j=1}^3\big) =& \big[\check{\mathbf{R}}_{L,1}(u_1, u_2)\check{\mathbf{R}}_{1,2}(u_1, u_3)\big]\big[\check{\mathbf{R}}_{3,4}(u_1, u_2)\check{\mathbf{R}}_{4,5}(u_1, u_3)\big]\cdots \\
& \times \cdots \big[\check{\mathbf{R}}_{L-3,L-2}(u_1, u_2)\check{\mathbf{R}}_{L-2,L-1}(u_1, u_3)\big], \\
\mathbf{V}_1\big(\{u_j\}_{j=1}^3\big) =& \big[\check{\mathbf{R}}_{L-1,L}(u_1, u_2)\check{\mathbf{R}}_{L,1}(u_1, u_3)\big]\big[\check{\mathbf{R}}_{2,3}(u_1, u_2)\check{\mathbf{R}}_{3,4}(u_1, u_3)\big]\cdots \\
& \times \cdots \big[\check{\mathbf{R}}_{L-3,L-2}(u_1, u_2)\check{\mathbf{R}}_{L-2,L-1}(u_1, u_3)\big].
\end{aligned}
\tag{4.4}
$$

It is easy to generalise to higher periods $n \geq 4$ following these two examples.

*Proof.* We start with defining the operator $\mathbf{W}(\{u_j\}_{j=1}^n)$ acting on the physical Hilbert space $(\mathbb{C}^N)^{\otimes L}$,

$$
\mathbf{W}\big(\{u_j\}_{j=1}^n\big) = \Big[ \prod_{m=1}^{L/n} \prod_{j=1}^{n-1} \check{\mathbf{R}}_{n(m-1)+j,n(m-1)+j+1}(u_1, u_{j+1})\Big]\mathbf{G}^{-1} = \mathbf{V}_n\big(\{u_j\}_{j=1}^n\big)\mathbf{G}^{-1}, \tag{4.5}
$$

where the right translational operator $\mathbf{G}$ is expressed in terms of permutation operator $\mathbf{P}_{a,b}$ as

$$
\mathbf{G} = \prod_{m=1}^{L-1} \mathbf{P}_{m,m+1}, \quad \mathbf{G}^{-1} = \prod_{m=L-1}^{1} \mathbf{P}_{m,m+1}. \tag{4.6}
$$

Taking into account Eq. (2.4) we immediately see that $\mathbf{G}$ translates an arbitrary operator $\mathbf{F}_m$ to the right by one site:

$$
\mathbf{G}\mathbf{F}_m\mathbf{G}^{-1} = \mathbf{F}_{m+1} \tag{4.7}
$$

Similar to the way how we obtain the transfer matrix from the monodromy matrix, we rewrite the operator $\mathbf{W}(\{u_j\}_{j=1}^n)$ as

$$
\mathbf{W}\big(\{u_j\}_{j=1}^n\big) = \mathrm{Tr}_b\, \tilde{\mathbf{W}}_b\big(\{u_j\}_{j=1}^n\big), \tag{4.8}
$$

$$
\tilde{\mathbf{W}}_b\big(\{u_j\}_{j=1}^n\big) = \prod_{m=1}^{L/n} \left[\mathbf{P}_{b,n(m-1)+1} \prod_{j=1}^{n-1} \mathbf{R}_{a,n(m-1)+j+1}(u_1, u_{j+1})\right]. \tag{4.9}
$$

Here we have used the properties of the permutation operator together with the "train trick" [95], i.e.

$$
\mathbf{P}_{a,b} = \mathbf{P}_{b,a}, \quad \mathbf{P}_{a,b}\mathbf{P}_{a,c} = \mathbf{P}_{b,c}\mathbf{P}_{a,b}, \quad \mathrm{Tr}_a\,\mathbf{P}_{a,b} = \mathbb{1}_b. \tag{4.10}
$$

The properties of the permutation operator (4.10) also imply that

$$
\mathbf{R}_{a,b}(u, u_1)\mathbf{R}_{a,m}(u, u_1)\mathbf{P}_{b,m} = \mathbf{P}_{b,m}\mathbf{R}_{a,m}(u, u_1)\mathbf{R}_{a,b}(u, u_1). \tag{4.11}
$$

Together with the Yang–Baxter equation (2.1), we realise that $\mathbf{R}_{a,b}(u, u_1)$ is essentially the intertwiner between the inhomogeneous monodromy matrix $\mathbf{M}_a\big(u, \{u_j\}_{j=1}^n\big)$ and $\tilde{\mathbf{W}}_b\big(\{u_j\}_{j=1}^n\big)$,

$$\mathbf{R}_{a,b}(u, u_1)\mathbf{M}_a\big(u, \{u_j\}_{j=1}^n\big)\tilde{\mathbf{W}}_b\big(\{u_j\}_{j=1}^n\big) = \tilde{\mathbf{W}}_b\big(\{u_j\}_{j=1}^n\big)\mathbf{M}_a\big(u, \{u_j\}_{j=1}^n\big)\mathbf{R}_{a,b}(u, u_1), \quad (4.12)$$

which implies the commutation relation

$$\big[\mathbf{T}\big(u, \{u_j\}_{j=1}^n\big), \mathbf{W}\big(\{u_j\}_{j=1}^n\big)\big] = 0, \quad \forall u \in \mathbb{C}. \quad (4.13)$$

Moreover, using the right translational operator $\mathbf{G}$, we have

$$\mathbf{V}_m\big(\{u_j\}_{j=1}^n\big) = \mathbf{G}^{m-p}\mathbf{V}_p\big(\{u_j\}_{j=1}^n\big)\mathbf{G}^{p-m}, \quad (4.14)$$

with $m, p \in \{1, 2, \cdots n\}$. We can therefore express the periodic Floquet evolution operator with period $n$ [see Eq. (4.1)] as

$$\mathbf{U}_{\mathrm{F}}\big(\{u_j\}_{j=1}^n\big) = \prod_{k=n}^{1} \mathbf{V}_k\big(\{u_j\}_{j=1}^n\big) = \mathbf{W}^n\big(\{u_j\}_{j=1}^n\big)\mathbf{G}^n. \quad (4.15)$$

Since the inhomogeneous transfer matrix commutes with $\mathbf{G}^n$ [6], it also commute with the periodic Floquet evolution operator with period $n$,

$$\big[\mathbf{U}_{\mathrm{F}}\big(\{u_j\}_{j=1}^n\big), \mathbf{T}\big(u, \{u_j\}_{j=1}^n\big)\big] = 0, \forall u \in \mathbb{C}. \quad (4.16)$$

$\square$

From this construction, we define the local Floquet quantum gate

$$\mathcal{U}_{m,m+n-1}\big(\{u_j\}_{j=1}^n\big) = \prod_{j=1}^{n-1} \check{\mathbf{R}}_{m+j-1,m+j}(u_1, u_{j+1}), \quad (4.17)$$

acting on the physical Hilbert space from $m$th site to $(m + n - 1)$th site. The Floquet evolution operator can be thus expressed as

$$\mathbf{U}_{\mathrm{F}}\big(\{u_j\}_{j=1}^n\big) = \prod_{k=n}^{1} \prod_{m=1}^{L/n} \mathcal{U}_{n(m-1)+k,nm+k-1}\big(\{u_j\}_{j=1}^n\big). \quad (4.18)$$

The relation to the Floquet dynamics is shown as follows. We define a Hamiltonian with periodic boundary conditions,

$$\mathbf{H}^{(n)} = \sum_{m=1}^{L} \mathbf{h}_{m,m+1,\cdots m+n-1}, \quad \mathbf{h}_{m,m+1,\cdots m+n-1} = \frac{\mathrm{i}}{t_0} \log \mathcal{U}_{m,m+n-1}, \quad t_0 \in \mathbb{R}, \quad (4.19)$$

such that

$$\mathcal{U}_{m,m+n-1} = \exp\big(-\mathrm{i}t_0 \mathbf{h}_{m,m+1,\cdots m+n-1}\big). \quad (4.20)$$

Then, we divide the Hamiltonian into $n$ parts,

$$\mathbf{H}_j^{(n)} = \sum_{m=1}^{L/n} \mathbf{h}_{n(m-1)+j+1,n(m-1)+j+2,\cdots nm+j}, \quad \mathbf{H}^{(n)} = \sum_{j=1}^{n} \mathbf{H}_j^{(n)}, \quad (4.21)$$

---

[6]The period in terms of the lattice sites of the inhomogeneous transfer matrix is exactly $n$, hence commuting with $\mathbf{G}^n$,

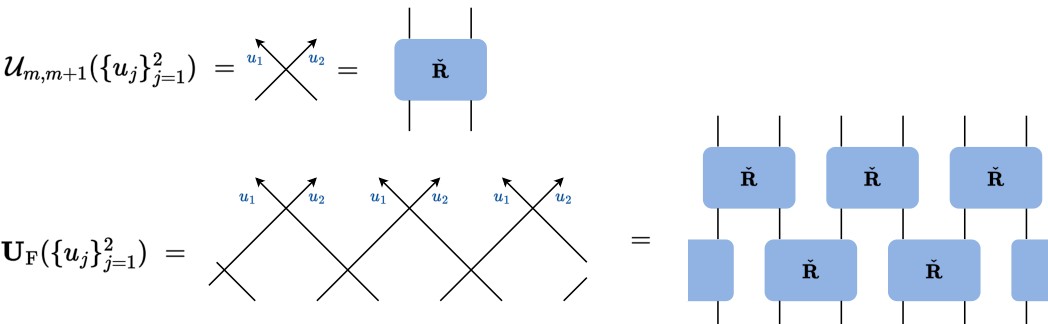

Figure 4: Diagrammatic demonstration of the Floquet evolution operator with period 2 as a tilted transfer matrix, the so-called "light-cone transfer matrix".

and the Floquet evolution operator becomes

$$\mathbf{U}_{\mathrm{F}}(\{u_j\}_{j=1}^n) = \prod_{j=n}^{1} \exp\left(-\mathrm{i}\mathbf{H}_j^{(n)}t_0\right),\qquad(4.22)$$

describing a Floquet dynamics with Floquet period $T = nt_0$. Note that even though the Floquet evolution operator is integrable, the Hamiltonian $\mathbf{H}^{(n)}$ might not be integrable itself.

Using the graphic representation of the R matrix, we can view the Floquet evolution operator (4.1) as a tilted transfer matrix of the underlying vertex model in a cylinder [13, 14, 94], as shown in Fig. 4. Note that the tilted transfer matrices do not commute with each other generally, i.e.

$$\left[\mathbf{U}_{\mathrm{F}}(\{u_j\}_{j=1}^n), \mathbf{U}_{\mathrm{F}}(\{v_j\}_{j=1}^n)\right] \neq 0,\qquad(4.23)$$

when $\{u_j\}_{j=1}^n$ and $\{v_j\}_{j=1}^n$ do not coincide.

The case of $n = 2$ has been studied in many previous works [13–15]. The cases with depth $n \geq 3$ can be obtained analogously, resulting in a different tilted transfer matrix, as illustrated in Figs. 5 and 6 for $n = 3$. We note that the tilted transfer matrices of depth $n \geq 3$ for the integrable vertex models might be useful in the context of thermodynamic limit of integrable vertex models with different geometries/topologies. Another observation is that the tilted transfer matrices with depth $n \geq 3$ are not "symmetric" with respect to the number of left and right directed lines, cf. Fig. 6. We also note that one can similarly obtain the integrable Floquet evolution operator with local terms

$$\tilde{\mathcal{U}}_{m,m+n-1} = \check{\mathbf{R}}_{m+n-1,m+n-2}(u_n, u_{n-1})\cdots\check{\mathbf{R}}_{m+2,m+1}(u_n, u_2)\check{\mathbf{R}}_{m+1,m}(u_n, u_1),\qquad(4.24)$$

by the same procedure of the Floquet Baxterisation. The relation between the two "chiral" and "anti-chiral" tilted transfer matrices (Floquet evolution operators) is postponed to future investigation.

**Remark.** This construction for integrable Floquet dynamics (or integrable quantum circuits) with depth $n \geq 3$ is new. It generalises the known results for the integrable Floquet dynamics of depth $n = 2$. The construction is different from the recent results on the medium-range integrable spin chain, where one can construct integrable Floquet dynamics of depth $n = 3$ [19]. However, the operator $\mathcal{U}_{m,m+2}$ cannot be decomposed into the product of two $\check{\mathbf{R}}$ operators, as discussed in this section. We think that there are different mechanisms to construct integrable Floquet dynamics of period $n \geq 3$, within which our method is not exhaustive.

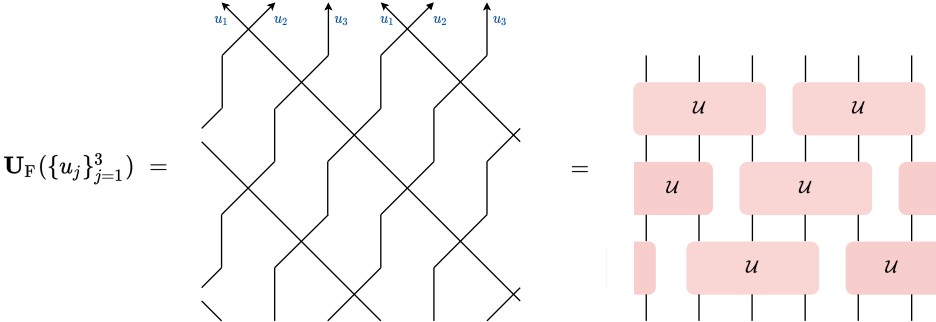

Figure 5: Diagrammatic demonstration of the local gate $\mathcal{U}_{m,m+2}$ with period 3.

Figure 6: Diagrammatic picture of the Floquet evolution operator with period 3 as a tilted transfer matrix.

## 4.2  Floquet Baxterisation for regular R matrix

After presenting the results for the generic case, where the regularity condition for the R matrix (2.10) is not assumed, for completeness we present the results for the scenario when the R matrix is regular, which has been studied previously [15, 17].

If the R matrix is regular ($\mathbf{R}_{a,b}(0,0) = \mathbf{P}_{a,b}$) and the inhomogeneity $u_1 = 0$, the Floquet evolution operator can be expressed directly in terms of the inhomogeneous transfer matrix, i.e.

$$
\begin{aligned}
\mathbf{W}\big(\{u_j\}_{j=1}^n\big)\big|_{u_1=0} &= \mathbf{T}\big(0, \{u_j\}_{j=1}^n\big)\big|_{u_1=0}\,, \\
\mathbf{U}_{\mathrm{F}}\big(\{u_j\}_{j=1}^n\big)\big|_{u_1=0} &= \mathbf{T}^n\big(0, \{u_j\}_{j=1}^n\big)\big|_{u_1=0}\,\mathbf{G}^n.
\end{aligned}
\tag{4.25}
$$

This relation is useful, since we obtain the complete spectrum of the Floquet evolution operator in terms of the eigenvalues of the inhomogeneous transfer matrix, which in turn can be expressed in terms of the solutions to the Bethe equations, quantum numbers that label the eigenstates of integrable transfer matrices [7]. This allows us to use thermodynamic Bethe ansatz to study the behaviour of the spectra of the Floquet evolution operators in the thermodynamic limit. A diagrammatic demonstration of the relation between the Floquet evolution operator and the inhomogeneous transfer matrix with $n = 2$ is shown in Fig. 7.

## 4.3  Floquet integrability with reflecting ends

In addition to the Floquet Baxterisation with periodic boundary condition presented in Sec. 4.1, we generalise the construction to the case with reflecting ends (for depth $n = 2$), i.e. open boundary condition, using the boundary Yang–Baxter equations. The boundary Yang–Baxter equations only apply when the R matrix satisfies the following properties [96, 97]: (1) the R matrix is of difference form, cf. (2.11); (2) the R matrix is regular, cf. (2.10); (3) the R matrix has the inversion and crossing symmetries, explained

---

[7]We explain the procedure with the example of the 6-vertex model in Sec. 5.

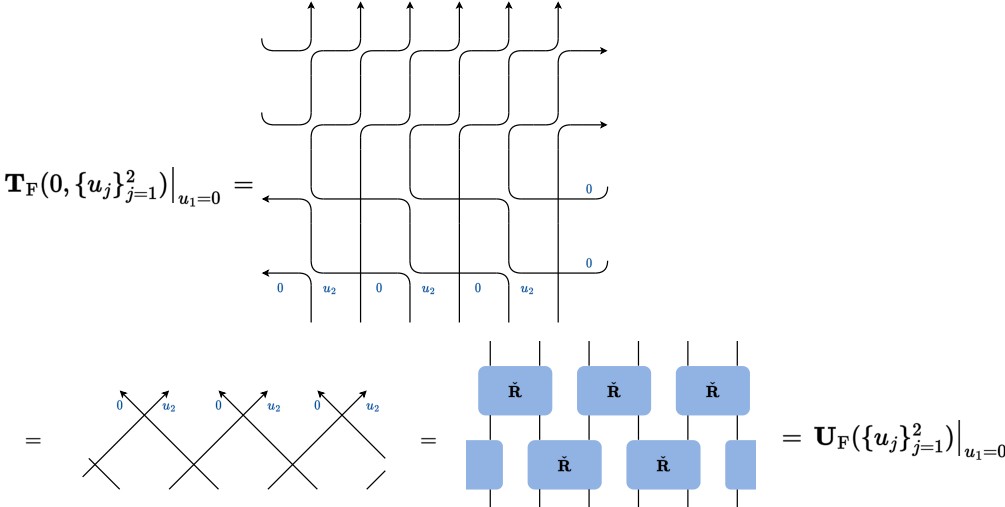

Figure 7: Diagrammatic demonstration of the relation between the Floquet evolution operator and the inhomogeneous transfer matrix with $n = 2$ when the R matrix is regular.

below. We assume the R matrix satisfies all these properties. Since the R matrix is of difference form, we denote the R matrix as $\mathbf{R}_{a,b}(u)$. The system size $L$ is assumed to be even to accommodate the depth of the Floquet evolution operator $n = 2$.

We begin with defining the inversion and crossing symmetries of the R matrix. The inversion symmetry of the R matrix means that

$$\mathbf{R}_{a,b}(u)\mathbf{R}_{a,b}^{t}(-u) = \mathbb{1}, \tag{4.26}$$

where $\mathbf{R}_{a,b}^{t} := \mathbf{R}_{a,b}^{t_a,t_b}$, i.e. the transpose of the R matrix in both spaces $a$ and $b$. Meanwhile, the crossing symmetry of the R matrix implies the existence of a constant operator $\mathbf{v}_a$ acting on space $a$ such that

$$\mathbf{R}_{a,b}(u) \propto \mathbf{v}_a \mathbf{R}_{a,b}^{t_b}(-u - \eta)\mathbf{v}_a^{-1}. \tag{4.27}$$

The crossing symmetry (4.27) can be equivalently expressed as

$$\mathbf{R}_{a,b}^{t_a}(u)\mathbf{w}_a \mathbf{R}_{a,b}^{t_b}(-u - 2\eta)\mathbf{w}_a^{-1} \propto \mathbb{1}, \tag{4.28}$$

where the operator $\mathbf{w}_a$ is

$$\mathbf{w}_a = \mathbf{v}_a^t \mathbf{v}_a. \tag{4.29}$$

When all three properties are satisfied, the boundary Yang–Baxter equations [96, 97] read

$$\mathbf{R}_{a,b}(u - v)\mathbf{K}_{-,a}(u)\mathbf{R}_{b,a}(u + v)\mathbf{K}_{-,b}(v) = \mathbf{K}_{-,b}(v)\mathbf{R}_{a,b}(u + v)\mathbf{K}_{-,a}(u)\mathbf{R}_{b,a}(u - v), \tag{4.30}$$

$$\begin{aligned}&\mathbf{R}_{a,b}(-u + v)\mathbf{K}_{+,a}^{t_a}(u)\mathbf{w}_a^{-1}\mathbf{R}_{a,b}^{t}(-u - v - 2\eta)\mathbf{w}_a \mathbf{K}_{+,b}^{t_b}(v) \\ &= \mathbf{K}_{+,b}^{t_b}(v)\mathbf{w}_a \mathbf{R}_{a,b}(-u - v - 2\eta)\mathbf{w}_a^{-1}\mathbf{K}_{+,a}^{t_a}(u)\mathbf{R}_{a,b}^{t}(-u + v).\end{aligned} \tag{4.31}$$

Using the boundary Yang–Baxter equations, we define the inhomogeneous monodromy matrix of period $n = 2$ with open boundaries (reflecting ends)

$$\mathbf{M}_a^{\mathrm{o}}\big(u, \{u_j\}_{j=1}^2\big) = \mathbf{M}_a\big(u, \{u_j\}_{j=1}^2\big)\mathbf{K}_{-,a}(u)\mathbf{M}_a^{-1}\big(-u, \{u_j\}_{j=1}^2\big)\mathbf{K}_{+,a}(u), \tag{4.32}$$

where after using the inversion symmetry (4.26) we obtain

$$
\mathbf{M}_a^{-1}\big(-u, \{u_j\}_{j=1}^2\big) = \mathbf{R}_{a,L}^t(u+u_2)\mathbf{R}_{a,L-1}^t(u+u_1)\cdots\mathbf{R}_{a,1}^t(u+u_1)
$$
$$
= \prod_{m=L/2}^{1} \mathbf{R}_{a,2m}^t(u+u_2)\mathbf{R}_{a,2m-1}^t(u+u_1). \tag{4.33}
$$

The inhomogeneous transfer matrix with open boundary is obtained by tracing over the auxiliary space of the monodromy matrix,

$$
\mathbf{T}^{\mathrm{o}}\big(u, \{u_j\}_{j=1}^2\big) = \mathrm{Tr}_a\, \mathbf{M}_a^{\mathrm{o}}\big(u, \{u_j\}_{j=1}^2\big). \tag{4.34}
$$

Using boundary Yang–Baxter equations (4.30) and (4.31), the inhomogeneous transfer matrices with open boundary condition are thus in involution,

$$
\big[\mathbf{T}^{\mathrm{o}}\big(u, \{u_j\}_{j=1}^2\big), \mathbf{T}^{\mathrm{o}}\big(v, \{u_j\}_{j=1}^2\big)\big] = 0, \quad \forall u, v \in \mathbb{C}. \tag{4.35}
$$

**Theorem 2.** *The Floquet time evolution operator with open boundary condition with system size $L \bmod 2 = 0$ is defined as*

$$
\mathbf{U}_{\mathrm{F}}^{\mathrm{o}}(\alpha) = \prod_{m=1}^{L/2} \check{\mathbf{R}}_{2m-1,2m}(-\alpha)\mathbf{K}_{-,L}\big(\tfrac{\alpha}{2}\big) \prod_{m=1}^{L/2-1} \check{\mathbf{R}}_{2m,2m+1}^t(-\alpha)\tilde{\mathbf{K}}_{+,1}\big(\tfrac{\alpha}{2}\big), \tag{4.36}
$$

*where*

$$
\tilde{\mathbf{K}}_{+,1}\big(\tfrac{\alpha}{2}\big) = \mathrm{Tr}_a\left[\mathbf{R}_{a,1}^t(-\alpha)\mathbf{K}_{+,a}\big(\tfrac{\alpha}{2}\big)\right]. \tag{4.37}
$$

*The Floquet time evolution operator with open boundary condition is integrable, i.e.*

$$
\left[\mathbf{U}_{\mathrm{F}}^{\mathrm{o}}(\alpha), \mathbf{T}^{\mathrm{o}}\big(u, \{\tfrac{\alpha}{2}, -\tfrac{\alpha}{2}\}\big)\right] = 0, \quad \forall u \in \mathbb{C}. \tag{4.38}
$$

*Proof.* To begin with, we express the transpose of the R matrix in terms of $\check{\mathbf{R}}$,

$$
\mathbf{R}_{a,b}^t(u) = \mathbf{P}_{a,b}\check{\mathbf{R}}_{a,b}^t(u). \tag{4.39}
$$

We consider the inhomogeneous transfer matrix with open boundary at $u = u_1 = -u_2 = \frac{\alpha}{2}$, where the constant $\alpha \in \mathbb{C}$,

$$
\mathbf{T}^{\mathrm{o}}\left(\tfrac{\alpha}{2}, \{\tfrac{\alpha}{2}, -\tfrac{\alpha}{2}\}\right) = \mathrm{Tr}_a\Bigg[\prod_{m=1}^{L/2}(\mathbf{P}_{a,2m-1}\check{\mathbf{R}}_{a,2m}(-\alpha)\mathbf{P}_{a,2m})\mathbf{K}_{-,a}\big(\tfrac{\alpha}{2}\big)
$$
$$
\prod_{m=L/2}^{1}(\mathbf{P}_{a,2m}\mathbf{P}_{a,2m-1}\check{\mathbf{R}}_{a,2m-1}^t(-\alpha))\mathbf{K}_{+,a}\big(\tfrac{\alpha}{2}\big)\Bigg]. \tag{4.40}
$$

For the first part of the expression, we move all the permutation operators to the right, i.e.

$$
\prod_{m=1}^{L/2}(\mathbf{P}_{a,2m-1}\check{\mathbf{R}}_{a,2m}(-\alpha)\mathbf{P}_{a,2m})\mathbf{K}_{-,a}\big(\tfrac{\alpha}{2}\big) = \prod_{m=1}^{L/2}\check{\mathbf{R}}_{2m-1,2m}(-\alpha)\mathbf{K}_{-,L}\big(\tfrac{\alpha}{2}\big)\mathbf{G}^{-1}\mathbf{P}_{a,1}, \tag{4.41}
$$

while for the remaining part, we move all the permutation operators to the left,

$$
\prod_{m=L/2}^{1}(\mathbf{P}_{a,2m}\mathbf{P}_{a,2m-1}\check{\mathbf{R}}_{a,2m-1}^t(-\alpha))\mathbf{K}_{+,a}\big(\tfrac{\alpha}{2}\big) = \mathbf{P}_{a,1}\mathbf{G}\prod_{m=1}^{L/2-1}\check{\mathbf{R}}_{2m,2m+1}^t(-\alpha)
$$
$$
\check{\mathbf{R}}_{a,1}^t(-\alpha)\mathbf{K}_{+,a}\big(\tfrac{\alpha}{2}\big). \tag{4.42}
$$

$$\mathbf{T}^{\mathrm{o}}\big(\tfrac{\alpha}{2},\{\tfrac{\alpha}{2},-\tfrac{\alpha}{2}\}\big) \;=\;$$ 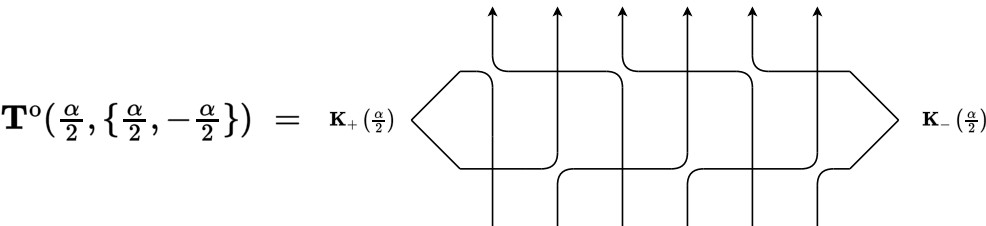

$$=$$ 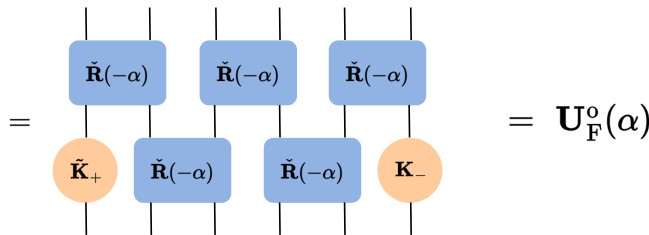 $$\;=\; \mathbf{U}_{\mathrm{F}}^{\mathrm{o}}(\alpha)$$

Figure 8: Diagrammatic demonstration of the Floquet evolution operator $\mathbf{U}_{\mathrm{F}}^{\mathrm{o}}(\alpha)$.

After multiplying the two parts, we observe that the dependence of the auxiliary space only exists on the last two terms, with which we could easily perform the operation of taking trace in the auxiliary space, i.e.

$$\mathrm{Tr}_a\left[\check{\mathbf{R}}_{a,1}^t(-\alpha)\mathbf{K}_{+,a}\big(\tfrac{\alpha}{2}\big)\right] = \tilde{\mathbf{K}}_{+,1}\big(\tfrac{\alpha}{2}\big). \tag{4.43}$$

Combining all the steps, we arrive at the following identity:

$$\mathbf{T}^{\mathrm{o}}\left(\tfrac{\alpha}{2},\{\tfrac{\alpha}{2},-\tfrac{\alpha}{2}\}\right) = \prod_{m=1}^{L/2}\check{\mathbf{R}}_{2m-1,2m}(-\alpha)\mathbf{K}_{-,L}\big(\tfrac{\alpha}{2}\big)\prod_{m=1}^{L/2-1}\check{\mathbf{R}}_{2m,2m+1}^t(-\alpha)\tilde{\mathbf{K}}_{+,1}\big(\tfrac{\alpha}{2}\big)$$
$$= \mathbf{U}_{\mathrm{F}}^{\mathrm{o}}(\alpha), \tag{4.44}$$

which means that the Floquet evolution operator with open boundary condition is precisely the inhomogeneous transfer matrix with open boundary condition with specific choices of the spectral parameter and inhomogeneities.

Therefore, one has

$$\left[\mathbf{U}_{\mathrm{F}}^{\mathrm{o}}(\alpha),\mathbf{T}^{\mathrm{o}}\left(u,\{\tfrac{\alpha}{2},-\tfrac{\alpha}{2}\}\right)\right] = \left[\mathbf{T}^{\mathrm{o}}\left(\tfrac{\alpha}{2},\{\tfrac{\alpha}{2},-\tfrac{\alpha}{2}\}\right),\mathbf{T}^{\mathrm{o}}\left(u,\{\tfrac{\alpha}{2},-\tfrac{\alpha}{2}\}\right)\right] = 0, \tag{4.45}$$

valid for any $u \in \mathbb{C}$, as a consequence of the boundary Yang–Baxter equations.

$$\square$$

We can visualise the relation between the Floquet evolution operator with open boundary condition and the inhomogeneous transfer matrix in Fig. 8.

The Floquet evolution operator with open boundary condition can be used to (partially) prove the conjectured integrability of a certain class of Temperley–Lieb algebraic Floquet protocols, which we will elaborate more on in Sec. 5 with the representation of the Temperley–Lieb (TL) algebra being the 6-vertex model.

# 5 Example: Floquet integrability in the staggered 6-vertex model

## 5.1 The affine Temperley–Lieb algebra

Before discussing integrability of the staggered 6-vertex model, we introduce the affine Temperley–Lieb (aTL) algebra [2, 31]. As we will see shortly, the Lax operator of the staggered 6-vertex model can be expressed in terms of a representation of the aTL algebra.

The aTL algebra is a unital associated algebra with the generators $g$ and $\{e_m | m \in \{1, 2, \cdots L-1\}\}$ that satisfy the relations:

$$e_m^2 = \beta e_m, \quad e_m e_{m\pm 1} e_m = e_m, \quad g e_m g^{-1} = e_{m+1}, \tag{5.1}$$

where $e_m = e_{m \bmod L}$. The aTL algebra is of great use in statistical mechanics, closely related to various loop and vertex models [2, 98, 99].

Recently, the authors of [26] conjectured the integrability criterion for any Floquet protocol that can be written as a representation of the (affine) Temperley–Lieb algebra. Using the construction of Floquet Baxterisation discussed in Sec. 4, here we can readily prove some of the conjectures given in [26].

We consider a representation of the aTL algebra on the Hilbert space $(\mathbb{C}^N)^{\otimes L}$,

$$e_m \to \mathbf{e}_{m,m+1}, \quad g \to \mathbf{G}, \tag{5.2}$$

where $\mathbf{e}_{m,m+1}$ acts locally on the $m$th and $(m+1)$th sites. From the aTL algebra, we have the following R matrix,

$$\check{\mathbf{R}}_{m,m+1}(u) = \mathbb{1} - \frac{\sinh(u)}{\sinh(u+\eta)} \mathbf{e}_{m,m+1}, \quad \mathbf{R}_{m,m+1}(u) = \check{\mathbf{R}}_{m,m+1}(u)\mathbf{P}_{m,m+1}. \tag{5.3}$$

where $\beta = 2\cosh\eta$. The R matrix (5.3) satisfies the Yang–Baxter relation (2.1) of difference form,

$$\check{\mathbf{R}}_{m,m+1}(u-v)\check{\mathbf{R}}_{m+1,m+2}(u)\check{\mathbf{R}}_{m,m+1}(v) = \check{\mathbf{R}}_{m+1,m+2}(v)\check{\mathbf{R}}_{m,m+1}(u)\check{\mathbf{R}}_{m+1,m+2}(u-v). \tag{5.4}$$

Moreover, the R matrix can be put into the exponential form using the properties (5.1), i.e.

$$\check{\mathbf{R}}_{m,m+1}(-\alpha) = \exp\left(-\mathrm{i}T\mathbf{e}_{m,m+1}\right) = \mathbb{1} + \frac{1}{\beta}\Big(\exp(\mathrm{i}\beta T) - 1\Big)\mathbf{e}_{m,m+1}, \tag{5.5}$$

where the relation between $u$ and $T$ is

$$\alpha = -\frac{1}{2}\log\left(\frac{\cosh(\eta - \mathrm{i}\cosh\eta T)}{\cosh(\eta + \mathrm{i}\cosh\eta T)}\right) + \mathrm{i}m\pi, \quad m \in \mathbb{Z}. \tag{5.6}$$

Then, using the results of Theorem 1 we construct the integrable Floquet evolution operator with period $n = 2$ in terms of the aTL generators, which reads

$$\begin{aligned} \mathbf{U}_{\mathrm{F}}(\{0,\alpha\}) &= \prod_{m=1}^{L/2} \check{\mathbf{R}}_{2m-1,2m}(-\alpha) \prod_{m=1}^{L/2} \check{\mathbf{R}}_{2m,2m+1}(-\alpha) \\ &= \prod_{m=1}^{L/2} \exp\left(-\mathrm{i}T\mathbf{e}_{2m-1,2m}\right) \prod_{m=1}^{L/2} \exp\left(-\mathrm{i}T\mathbf{e}_{2m,2m+1}\right), \end{aligned} \tag{5.7}$$

proving partially the conjecture in [26].

We discuss the case of open boundary condition (which requires considering representations of the usual Temperley–Lieb algebra instead of the affine one) in Sec. 5.3.

## 5.2 The staggered 6-vertex model

From now on, let us focus on a specific representation of the aTL algebra, where the underlying integrability structure of the Floquet evolution operator (5.7) with depth $n = 2$ is the staggered 6-vertex model (with even system size $L$). In this case, we introduce the operators $\mathbf{e}_{m,n}$ acting on the Hilbert space $(\mathbb{C}^2)^{\otimes L}$ (with periodic boundary condition $\mathbf{e}_{m,L+1} = \mathbf{e}_{m,1}$)

$$\mathbf{e}_{m,n} = \frac{q + q^{-1}}{4} - \frac{1}{2}\left(\sigma_m^x \sigma_n^x + \sigma_m^y \sigma_n^y + \frac{q + q^{-1}}{2}\sigma_m^z \sigma_n^z\right) - \frac{q - q^{-1}}{4}(\sigma_m^z - \sigma_n^z), \qquad (5.8)$$

with $\beta = q + q^{-1} = 2\cosh\eta$, i.e. $q = \exp\eta$. One can easily check that the operators $\mathbf{e}_{m,m+1}$ provide a representation of the aTL algebra (5.1). We also need the operator $\mathbf{G}$, the right translation operator on $(\mathbb{C}^2)^{\otimes L}$,

$$\mathbf{G} = \prod_{m=1}^{L-1} \mathbf{P}_{m,m+1}, \quad \mathbf{P}_{a,b} = \frac{1}{2}\left(\vec{\sigma}_a \cdot \vec{\sigma}_b + \mathbb{1}\right). \qquad (5.9)$$

The quantum spin-1/2 XXZ model is expressed in terms of the aTL generators [98],

$$\mathbf{H}_{\text{XXZ}} = -\sum_{m=1}^{L} \mathbf{e}_{m,m+1}. \qquad (5.10)$$

From the representation of the aTL algebra (5.8), the R matrix of the 6-vertex model reads [2,98]

$$\check{\mathbf{R}}_{a,m}(u) = \mathbb{1}_{a,m} - \frac{\sinh u}{\sinh(u + \eta)}\mathbf{e}_{a,m},$$

$$\mathbf{R}_{a,m}(u) = \check{\mathbf{R}}_{a,m}(u)\mathbf{P}_{a,m}$$

$$= \frac{1}{\sinh(u + \eta)}\begin{pmatrix} \sinh(u + \eta) & 0 & 0 & 0 \\ 0 & \sinh u & e^u \sinh\eta & 0 \\ 0 & e^{-u}\sinh\eta & \sinh u & 0 \\ 0 & 0 & 0 & \sinh(u + \eta) \end{pmatrix}, \qquad (5.11)$$

satisfying Yang–Baxter relations (2.2), (2.1) and (2.5).

For the purpose of studying the properties of the Floquet evolution operator (5.7), we concentrate on the staggered monodromy matrix [8],

$$\mathbf{M}_a(u, \alpha) = \prod_{m=1}^{L/2} \mathbf{R}_{a,2m-1}(u)\mathbf{R}_{a,2m}(u - \alpha) = \begin{pmatrix} \mathbf{A}(u, \alpha) & \mathbf{B}(u, \alpha) \\ \mathbf{C}(u, \alpha) & \mathbf{D}(u, \alpha) \end{pmatrix}_a. \qquad (5.12)$$

The staggered transfer matrix is defined as

$$\mathbf{T}(u, \alpha) = [\sinh(u + \eta)\sinh(u - \alpha + \eta)]^{L/2}\,\text{Tr}_a\,\mathbf{M}_a(u, \alpha), \qquad (5.13)$$

where the scalar prefactor is included for later convenience. From the Yang–Baxter relation (2.2), we have

$$\mathbf{R}_{a,b}(u, v)\mathbf{M}_a(u)\mathbf{M}_b(v) = \mathbf{M}_b(v)\mathbf{M}_a(u)\mathbf{R}_{a,b}(u, v). \qquad (5.14)$$

Therefore, the staggered transfer matrices are in involution, i.e.

$$[\mathbf{T}(u, \alpha), \mathbf{T}(v, \alpha)] = 0, \quad \forall u, v \in \mathbb{C}. \qquad (5.15)$$

---

[8]We set the inhomogeneities $u_1 = 0$, $u_2 = \alpha$ without losing generality due to the properties of the R matrix (2.11)

In fact, the Yang–Baxter equation (2.2) implies that the relations between the "quantum operators" [ i.e. $\mathbf{A}(u, \alpha)$, $\mathbf{B}(u, \alpha)$, $\mathbf{C}(u, \alpha)$ and $\mathbf{D}(u, \alpha)$] are the same as for the algebraic Bethe ansatz in the homogeneous case.

In order to find the eigenvalues of the Floquet evolution operator, we need to obtain the complete spectrum of the staggered transfer matrix first. In fact, the (unnormalised) eigenstates of the staggered transfer matrix are labelled by the set of quantum numbers $\{u_m\}_{m=1}^M$, the so called Bethe roots or rapidities, as follows from the algebraic Bethe ansatz:

$$|\{u_m\}_{m=1}^M\rangle = \prod_{m=1}^M \mathbf{B}(u_m, \alpha)|\Uparrow\rangle, \tag{5.16}$$

where the ferromagnetic pseudo-vacuum is $|\Uparrow\rangle = |\uparrow\uparrow\cdots\rangle$. In addition, the Bethe roots are zeros of the eigenvalues of the Baxter's Q operator,

$$\mathbf{Q}(u)|\{u_m\}_{m=1}^M\rangle = \prod_{m=1}^M \sinh(u - u_m)|\{u_m\}_{m=1}^M\rangle, \tag{5.17}$$

where the Q operator satisfies the renowned TQ relation [2] with the staggered transfer matrix,

$$\mathbf{T}(u, \alpha)\mathbf{Q}(u) = \mathbf{T}_0(u + \eta)\mathbf{Q}(u - \eta) + \mathbf{T}_0(u)\mathbf{Q}(u + \eta), \tag{5.18}$$

where the scalar function $\mathbf{T}_0$ is

$$\mathbf{T}_0(u, \alpha) = \prod_{n=1}^L \sinh(u - \xi_n) = [\sinh(u)\sinh(u - \alpha)]^{L/2}. \tag{5.19}$$

The Q operator commutes with the staggered transfer matrix, and it can be constructed from the $\infty$-dimensional highest weight representation of the underlying quantum group $\mathcal{U}_q(\mathfrak{sl}_2)$. The inhomogeneous case can be obtained using the same factorisation properties of the transfer matrices as in [100], and similar simplification (truncation in the auxiliary space) works for the root of unity case. A more detailed discussion is postponed to future work.

By taking the limit $u \to u_m$, we obtain a set of non-linear equations that the Bethe roots must satisfy, i.e. the Bethe equations [3],

$$\left(\frac{\sinh(u_m + \eta)\sinh(u_m - \alpha + \eta)}{\sinh u_m \sinh(u_m - \alpha)}\right)^{L/2} = \prod_{n \neq m}^M \frac{\sinh(u_m - u_n + \eta)}{\sinh(u_m - u_n - \eta)}. \tag{5.20}$$

Defining $\lambda_m = u_m - \frac{\alpha}{2} + \frac{\eta}{2}$, we have

$$\left(\frac{\sinh(\lambda_m + \alpha/2 + \eta/2)\sinh(\lambda_m - \alpha/2 + \eta/2)}{\sinh(\lambda_m + \alpha/2 - \eta/2)\sinh(\lambda_m - \alpha/2 - \eta/2)}\right)^{L/2} = \prod_{n \neq m}^M \frac{\sinh(\lambda_m - \lambda_n + \eta)}{\sinh(\lambda_m - \lambda_n - \eta)}. \tag{5.21}$$

When $\alpha \in \mathbb{R}$, the values of $\lambda_m$ satisfy certain constraints, as shown in Appendix B.

When the value of $q$ is at root of unity ($q^n = 1$, $n \in \mathbf{Z}_+$), there exist solutions with Bethe roots belonging to the so called "exact strings", similar to the homogeneous case studied in [101–103]. In this case, the algebraic Bethe ansatz (ABA) for the staggered 6-vertex model becomes subtle. However, construction of the Q operator and the eigenstates with "exact strings" is very similar to the homogeneous case, which has been demonstrated in [100, 104]. We postpone the discussion on the details about the impact of the "exact strings" to future work. Meanwhile, the existence of the "exact strings" does not affect the

eigenvalues of the transfer matrix, hence the results of the present work remain correct even at root of unity values of $q$.

The eigenvalues of the staggered transfer matrix for any eigenstate are expressed in terms of the Bethe roots, i.e.

$$
\begin{aligned}
\mathbf{T}(u,\alpha)|\{u_m\}_{m=1}^M\rangle &= \left[\mathbf{T}_0(u+\eta)\mathbf{Q}(u-\eta) + \mathbf{T}_0(u)\mathbf{Q}(u+\eta)\right]\mathbf{Q}^{-1}(u)|\{u_m\}_{m=1}^M\rangle \\
&= \tau(u,\alpha,\{u_m\}_{m=1}^M)|\{u_m\}_{m=1}^M\rangle,
\end{aligned}
\tag{5.22}
$$

$$
\begin{aligned}
\tau(u,\alpha,\{u_m\}_{m=1}^M) &= \frac{1}{\prod_{m=1}^M \sinh(u-u_m)}\Big( [\sinh(u+\eta)\sinh(u-\alpha+\eta)]^{L/2} \\
&\prod_{m=1}^M \sinh(u-u_m-\eta) + [\sinh(u)\sinh(u-\alpha)]^{L/2}\prod_{m=1}^M \sinh(u-u_m+\eta)\Big).
\end{aligned}
\tag{5.23}
$$

Before we move to the spectrum of the Floquet evolution operator, we would like to take a look at the local conserved charges of the staggered 6-vertex model. In order to construct the conserved charges with a local density, we define a two-row transfer matrix $\tilde{\mathbf{T}}(u,\alpha)$,

$$
\tilde{\mathbf{T}}(u,\alpha) = \mathbf{T}(u,\alpha)\mathbf{T}(u+\alpha,\alpha).
\tag{5.24}
$$

The physical momentum is defined through

$$
\frac{\tilde{\mathbf{T}}(0,\alpha)}{\left[\sinh^2\eta \sinh(\eta-\alpha)\sinh(\eta+\alpha)\right]^{L/2}} = \mathbf{G}^{-2} = \exp(-\mathrm{i}\mathbf{p}).
\tag{5.25}
$$

For a Bethe state $|\{u_m\}_{m=1}^M\rangle$ one has

$$
\mathbf{G}^{-2}|\{u_m\}_{m=1}^M\rangle = \prod_{m=1}^M \frac{\sinh(u_m+\eta)\sinh(u_m-\alpha+\eta)}{\sinh(u_m)\sinh(u_m-\alpha)}|\{u_m\}_{m=1}^M\rangle.
\tag{5.26}
$$

Thus the momentum eigenvalue of the eigenstate $|\{u_m\}_{m=1}^M\rangle$ becomes

$$
\mathrm{i}\sum_{m=1}^M \log \frac{\sinh(u_m+\eta)\sinh(u_m-\alpha+\eta)}{\sinh(u_m)\sinh(u_m-\alpha)}.
\tag{5.27}
$$

We obtain the staggered Hamiltonian by taking the logarithmic derivative of the two-row transfer matrix

$$
\mathbf{H}_{\mathrm{st}} = \frac{\sinh\eta}{2}\partial_u \log \tilde{\mathbf{T}}(u,\alpha)\Big|_{u=0}.
\tag{5.28}
$$

The staggered Hamiltonian can be nicely expressed in terms of the TL generators, i.e.

$$
\begin{aligned}
\mathbf{H}_{\mathrm{st}} = -\sum_{m=1}^L \Bigg( &\mathbf{e}_m + \frac{\sinh^2\alpha\cosh\eta}{\cosh(2\eta)-\cosh(2\alpha)}\{\mathbf{e}_m,\mathbf{e}_{m+1}\} \\
&-(-1)^m \frac{\sinh\alpha\cosh\alpha\sinh\eta}{\cosh(2\eta)-\cosh(2\alpha)}[\mathbf{e}_m,\mathbf{e}_{m+1}] - \frac{\cosh\eta(\cosh(2\eta)-\cosh^2\alpha)}{\cosh(2\eta)-\cosh(2\alpha)}\Bigg),
\end{aligned}
\tag{5.29}
$$

where $\mathbf{e}_m := \mathbf{e}_{m,m+1}$. This staggered Hamiltonian coincides with the first non-trivial conserved quantity found in [26] [see Eq. (3.5)] with periodic boundary condition. It also precisely recovers the corresponding expression in [52] up to a constant. As discussed in Sec. 3.1, this Hamiltonian is nothing else that the lattice limit of the black hole sigma model (non-compact CFT), which we shall not repeat again.

## 5.3 Floquet time evolution operator: open boundary condition

When we consider the case with open boundary condition, the aTL algebra reduces the TL algebra [9], whose representation on $(\mathbb{C}^2)^{\otimes L}$ is the centralizer of the representation on the same space of the quantum group $\mathcal{U}_q(\mathfrak{sl}_2)$ [98] due to the quantum Schur–Weyl duality. Subsequently, this leads to the renowned quantum group invariant spin-1/2 XXZ model that can be studied through the representation theory of $\mathcal{U}_q(\mathfrak{sl}_2)$ [98,105]. In the context of the Floquet evolution operator with open boundary condition, we need to study the inhomogeneous transfer matrix of the staggered 6-vertex model with open boundary condition.

Since the R matrix satisfies the regularity (2.10), inversion (4.26) and crossing symmetries (4.27) and (4.28), the boundary Yang–Baxter relations also hold for the 6-vertex R matrix. We take a slight detour to present a special choice of the K matrices that correspond to the $U_q(\mathfrak{sl}_2)$-invariant Floquet evolution operator [98].

In the case of the R matrix in (5.11), for the crossing symmetries (4.27) and (4.28), we have

$$\mathbf{v}_a = \begin{pmatrix} 0 & e^{-\eta/2} \\ e^{\eta/2} & 0 \end{pmatrix}_a, \quad \mathbf{w}_a = \mathbf{v}_a^t \mathbf{v}_a = \begin{pmatrix} e^{\eta} & 0 \\ 0 & e^{-\eta} \end{pmatrix}_a. \tag{5.30}$$

Let us consider the $U_q(\mathfrak{sl}_2)$-invariant case. Then, the boundary K matrices that satisfy the boundary Yang–Baxter relations (4.30) and (4.31) are independent of the spectral parameter $u$, i.e.

$$\mathbf{K}_{+,a}(u) = \mathbf{K}_{+,a} = \mathbf{w}_a, \quad \mathbf{K}_{-,a}(u) = \mathbf{K}_{-,a} = \mathbb{1}_a. \tag{5.31}$$

Moreover,

$$\tilde{\mathbf{K}}_{+,1}(-\alpha/2) = \mathrm{Tr}_a \left( \check{\mathbf{R}}_{a,1}(-\alpha)\mathbf{K}_{+,a} \right) = \frac{\sinh(2\eta - \alpha)}{\sinh^2(\eta - \alpha)}. \tag{5.32}$$

Therefore, the $U_q(\mathfrak{sl}_2)$-invariant Floquet evolution operator (with open boundaries) becomes

$$\mathbf{U}_\mathrm{F}^\mathrm{o}(\alpha) = \sinh^{L-2}(\eta - \alpha)\sinh(2\eta - \alpha) \prod_{m=1}^{L/2} \check{\mathbf{R}}_{2m-1,2m}(-\alpha) \prod_{m=1}^{L/2-1} \check{\mathbf{R}}_{2m,2m+1}(-\alpha). \tag{5.33}$$

The $U_q(\mathfrak{sl}_2)$-invariant Floquet evolution operator coincides with the Floquet evolution operator in [26] when using the representation in (5.8). This result demonstrates a special case of the Floquet integrability associated with the TL algebra, conjectured in [26]. The corresponding staggered 6-vertex model with open boundary condition has been studied in [44], where the low-energy spectrum are discussed. The charges with local density found in [26] for the TL representation in Eq. (5.8) can be obtained by taking the logarithmic derivatives of the staggered transfer matrix with open boundary condition [44].

## 5.4 Spectrum of the Floquet evolution operator

We now return our focus to the Floquet evolution operator with periodic boundary condition. Using the properties (5.1) of the aTL generators, the Floquet evolution operator $\mathbf{U}_\mathrm{F}(T)$ can be written as the products of the exponentials of the aTL generators (5.7),

$$\mathbf{U}_\mathrm{F}(T) := \mathbf{U}_\mathrm{F}(\{0, \alpha\}) = \prod_{m=1}^{L/2} \exp\left(-\mathrm{i}T\mathbf{e}_{2m-1,2m}\right) \prod_{m=1}^{L/2} \exp\left(-\mathrm{i}T\mathbf{e}_{2m,2m+1}\right). \tag{5.34}$$

---

[9]In our case, the TL algebra can be easily obtained by neglecting the generators $e_L$ and $g$ in the aTL algebra.

This Floquet evolution operator $\mathbf{U}_{\mathrm{F}}(T)$ generates the Floquet stroboscopic time evolution of the following protocol with Floquet period $2T$,

$$\mathbf{H}(t) = \begin{cases} \mathbf{H}_1 = \sum_{m=1}^{L/2} \mathbf{e}_{2m-1,2m} & 0 \leq t < T, \\ \mathbf{H}_2 = \sum_{m=1}^{L/2} \mathbf{e}_{2m,2m+1} & T \leq t < 2T, \end{cases} \qquad \mathbf{H}(t + 2T) = \mathbf{H}(t). \tag{5.35}$$

It is physical to consider the Floquet period $2T \in \mathbb{R}_+$.

The relation between the spectral parameter $\alpha$ and the Floquet period $2T$ is given in (5.6). The Floquet Hamiltonian $\mathbf{H}_{\mathrm{F}}$ is the effective Hamiltonian of the Floquet evolution operator,

$$\begin{aligned} \mathbf{U}_{\mathrm{F}}(T) &= \prod_{m=1}^{L/2} \left( \mathbb{1} + \frac{\exp(\mathrm{i}\beta T) - 1}{\beta} \mathbf{e}_{2m-1,2m} \right) \prod_{m=1}^{L/2} \left( \mathbb{1} + \frac{\exp(\mathrm{i}\beta T) - 1}{\beta} \mathbf{e}_{2m,2m+1} \right) \\ &= \prod_{m=1}^{L/2} \check{\mathbf{R}}_{2m-1,2m}(-\alpha) \prod_{m=1}^{L/2} \check{\mathbf{R}}_{2m,2m+1}(-\alpha) \\ &= \exp\left( -2\mathrm{i}\mathbf{H}_{\mathrm{F}}(\alpha)T \right), \end{aligned} \tag{5.36}$$

with $\alpha = \alpha(T)$ as in (5.6). The extra factor of 2 is due to the fact that the total Floquet period is $T_{\mathrm{total}} = 2T$.

The relation between the Floquet evolution operator and the staggered 6-vertex transfer matrix is

$$\mathbf{U}_{\mathrm{F}}(T) = \frac{1}{[\sinh \eta \sinh(\eta - \alpha)]^{L/2}} \mathbf{T}^2(0, \{0, \alpha\})\mathbf{G}^2, \tag{5.37}$$

which allows us to obtain the eigenvalues of the Floquet evolution operator in terms of the Bethe roots using (5.22) and (5.26),

$$\begin{aligned} \mathbf{U}_{\mathrm{F}}(T)|\{u_m\}_{m=1}^M\rangle &= \prod_{m=1}^M \frac{\sinh(u_m + \eta)\sinh(u_m - \alpha)}{\sinh(u_m)\sinh(u_m - \alpha + \eta)}|\{u_m\}_{m=1}^M\rangle \\ &= \prod_{m=1}^M \frac{\sinh(\lambda_m + \alpha/2 + \eta/2)\sinh(\lambda_m - \alpha/2 - \eta/2)}{\sinh(\lambda_m + \alpha/2 - \eta/2)\sinh(\lambda_m - \alpha/2 + \eta/2)}|\{u_m\}_{m=1}^M\rangle. \end{aligned} \tag{5.38}$$

In the meantime, the Floquet Hamiltonian is

$$\begin{aligned} \mathbf{H}_{\mathrm{F}}(T) &= \frac{\mathrm{i}}{2T}\log \mathbf{U}_{\mathrm{F}}(T) \\ &= \frac{\mathrm{i}}{T}\log \frac{\mathbf{T}(0, \{0, \alpha\})}{[\sinh \eta \sinh(\eta - \alpha)]^{L/2}} - \frac{1}{2T}\mathbf{p}, \end{aligned} \tag{5.39}$$

where the momentum is defined in Eq. (5.25),

$$\mathbf{p} = -\mathrm{i}\log \mathbf{G}^2. \tag{5.40}$$

Therefore, the eigenvalues of the Floquet Hamiltonian become

$$\begin{aligned} \mathbf{H}_{\mathrm{F}}(T)|\{u_m\}_{m=1}^M\rangle &= \frac{\mathrm{i}}{2T}\sum_{m=1}^M \log \frac{\sinh(u_m + \eta)\sinh(u_m - \alpha)}{\sinh(u_m)\sinh(u_m - \alpha + \eta)}|\{u_m\}_{m=1}^M\rangle \\ &= \frac{\mathrm{i}}{2T}\sum_{m=1}^M \log \frac{\sinh(\lambda_m + \alpha/2 + \eta/2)\sinh(\lambda_m - \alpha/2 - \eta/2)}{\sinh(\lambda_m + \alpha/2 - \eta/2)\sinh(\lambda_m - \alpha/2 + \eta/2)}|\{u_m\}_{m=1}^M\rangle. \end{aligned} \tag{5.41}$$

In general, the Floquet Hamiltonian cannot be expressed in terms of local operators.

We now move on to study under what circumstance the eigenvalues of the Floquet Hamiltonian are real, or equivalently, the eigenvalues of the Floquet evolution operator locate on the unit circle.

We start with the easy-axis and isotropic regime ($\eta \in \mathbb{R}$ or $|\Delta| \geq 1$). In this case, the inhomogeneity parameter $\alpha \in i\mathbb{R}$ with $T \in \mathbb{R}_+$ by solving (5.6). The isotropic limit $|\Delta| = 1$ is discussed in Appendix C.

Using the properties of the aTL generators (5.1), we obtain

$$\check{\mathbf{R}}_{a,b}(\alpha)\check{\mathbf{R}}_{a,b}(-\alpha) = \check{\mathbf{R}}_{a,b}(-\alpha)\check{\mathbf{R}}_{a,b}(\alpha) = \mathbb{1}. \tag{5.42}$$

Moreover, since in the easy-axis and isotropic regime one has $\alpha \in \mathbb{R}$, and

$$\mathbf{e}_{a,b}^\dagger = \mathbf{e}_{a,b}, \quad \eta \in \mathbb{R}, \tag{5.43}$$

we immediately obtain

$$\check{\mathbf{R}}_{a,b}^\dagger(\alpha) = \check{\mathbf{R}}_{a,b}(-\alpha) = \check{\mathbf{R}}_{a,b}^{-1}(\alpha). \tag{5.44}$$

This implies that the Floquet evolution operator $\mathbf{U}_\mathrm{F}(\alpha)$ is unitary, i.e.

$$\mathbf{U}_\mathrm{F}^\dagger(T)\mathbf{U}_\mathrm{F}(T) = \mathbf{U}_\mathrm{F}(T)\mathbf{U}_\mathrm{F}^\dagger(T) = \mathbb{1}, \quad \eta \in \mathbb{R}. \tag{5.45}$$

Equivalently, in this case the Floquet Hamiltonian is Hermitian. i.e. has a real spectrum. Therefore, in the easy-axis and isotropic regime, the Floquet evolution operator generates a unitary time evolution, or a unitary quantum circuit.

It turns out that the situation is different for the easy-plane regime, i.e. $\eta \in i\mathbb{R}/\{0\}$ ($|\Delta| < 1$), as the Floquet evolution operator $\mathbf{U}_\mathrm{F}(T)$ is no longer unitary. Moreover, the eigenvalues of $\mathbf{H}_\mathrm{F}(T)$ acquire a non-zero imagianry part at a certain value of $T$, leading to a (dynamical) phase transition between the anti-unitary symmetric and anti-unitary symmetry broken phases, which are highlighted as follows.

In the easy-plane regime ($\eta \in i\mathbb{R}/\{0\}$ or $|\Delta| < 1$), the Floquet evolution operator $\mathbf{U}_\mathrm{F}(T)$ becomes non-unitary. The reason is that

$$\mathbf{e}_{a,b}^\dagger \neq \mathbf{e}_{a,b}, \quad \eta \in i\mathbb{R}/\{0\}, \tag{5.46}$$

and therefore

$$\mathbf{U}_\mathrm{F}^\dagger(T) \neq \mathbf{U}_\mathrm{F}(-T) = \mathbf{U}_\mathrm{F}^{-1}(T), \quad \eta \in i\mathbb{R}/\{0\}. \tag{5.47}$$

Since the Floquet evolution operator is not unitary, we do not expect the eigenvalues of the Floquet evolution operator to locate along the unit circle. Thus, let us take a closer look at the eigenvalues of the Floquet evolution operator in terms of Bethe roots (5.38). Using the results of Appendix B, one can show that $\lambda_n$ belongs to one of the following three categories:

- $\lambda_n \in \mathbb{R}$,

- $\mathrm{Im}\,\lambda_n = \frac{\pi}{2}$ ,

- $\{\lambda_m\}_{m=1}^M$ contains both $\lambda_n$ and $\bar{\lambda}_n$ ,

for $\eta \in i\mathbb{R}/\{0\}$ and $\alpha \in \mathbb{R}$. This implies that the eigenvalues of the Floquet evolution operator are uni-modular, i.e.

$$\left| \prod_{m=1}^M \frac{\sinh(\lambda_m + \alpha/2 + \eta/2)\sinh(\lambda_m - \alpha/2 - \eta/2)}{\sinh(\lambda_m + \alpha/2 - \eta/2)\sinh(\lambda_m - \alpha/2 + \eta/2)} \right| = 1, \qquad \eta \in i\mathbb{R}/\{0\},\ \alpha \in \mathbb{R}. \tag{5.48}$$

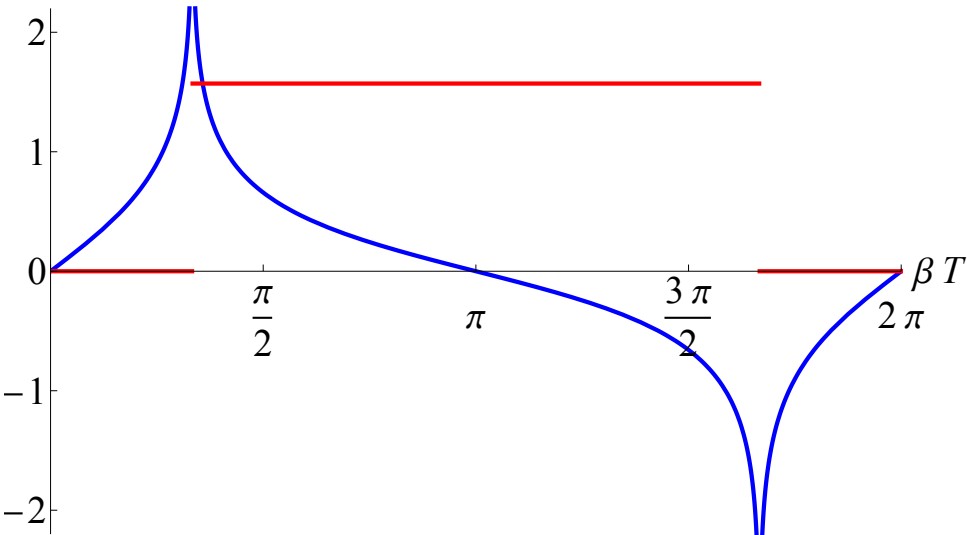

Figure 9: $\alpha$ as a function of $\beta T$ with $\eta = \mathrm{i}\pi/3$. The blue and red curves depict $\mathrm{Re}\,\alpha$ and $\mathrm{Im}\,\alpha$ changing with $\beta T$, respectively. Noted that $\mathrm{Re}\,\alpha$ diverges at $\beta T = \pi \pm 2\mathrm{i}\eta$.

Therefore, the eigenvalues of the Floquet Hamiltonian

$$\mathbf{H}_\mathrm{F}(\alpha) = \frac{\mathrm{i}\log\mathbf{U}_\mathrm{F}(\alpha)}{2T} \tag{5.49}$$

are real with $\eta \in \mathrm{i}\mathbb{R}/\{0\}$ and $\alpha \in \mathbb{R}$. In this case, the Floquet Hamiltonian can be transformed into a Hermitian Hamiltonian via a similarity transformation $\mathbf{x}$,

$$\mathbf{H}'_\mathrm{F}(\alpha) = \mathbf{x}\mathbf{H}_\mathrm{F}(\alpha)\mathbf{x}^{-1}, \quad \left(\mathbf{H}'_\mathrm{F}(\alpha)\right)^\dagger = \mathbf{H}'_\mathrm{F}(\alpha). \tag{5.50}$$

The Hermiticity condition of $\mathbf{H}'_\mathrm{F}(\alpha)$ can be rewritten as

$$\mathbf{X}\mathbf{H}_\mathrm{F}(\alpha) = \mathbf{H}^\dagger_\mathrm{F}(\alpha)\mathbf{X}, \quad \mathbf{X} = \mathbf{x}^\dagger\mathbf{x}, \tag{5.51}$$

where the Hermitian operator $\mathbf{X}$ is known as the Dyson map [106].

For any eigenstate of $\mathbf{H}_\mathrm{F}(\alpha)$ $|\psi\rangle$ with eigenvalue $E$, we have

$$\mathbf{x}\mathbf{H}_\mathrm{F}(\alpha)|\psi\rangle = E\mathbf{x}|\psi\rangle = \mathbf{H}'_\mathrm{F}(\alpha)\mathbf{x}|\psi\rangle. \tag{5.52}$$

Hence, $\mathbf{H}_\mathrm{F}(\alpha)$ has the same real spectrum as the Hermitian operator $\mathbf{H}'_\mathrm{F}(\alpha)$. Their eigenstates are related by the operator $\mathbf{x}$. When Hermitian operator $\mathbf{X}$ is positive and invertible, the Floquet Hamiltonian is pseudo/quasi-Hermitian [107]. From the Dyson map, we are able to define proper inner product for the left and right eigenstates of the Floquet Hamiltonian. A detailed discussion on this can be found in [108].

The exact form of the Dyson map $\mathbf{X}$ can be constructed from the (left and right) eigenstates of the Floquet Hamiltonian $\mathbf{H}_\mathrm{F}$. Moreover, when the Floquet Hamiltonian contains Jordan blocks after block-diagonalising the matrix, the Hermitian operator $\mathbf{X}$ contains further complications, as shown in [109]. We shall leave the investigation of the exact form of the Dyson map $\mathbf{X}$ to future work.

Above all, from (5.6) that relates the inhomogeneity $\alpha$ and the Floquet period $2T$ and Fig. 9, we observe that the imaginary part of $\alpha$ experience a $\frac{\pi}{2}$ jump, which implies a sudden change with respect to the spectra of the Floquet evolution operators.

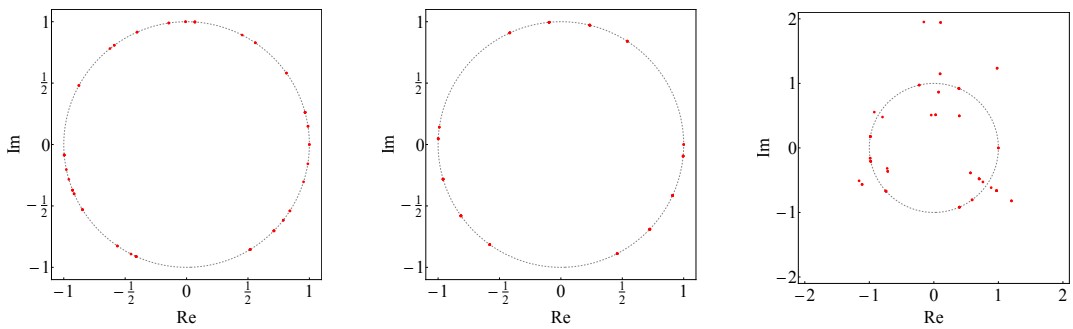

Figure 10: Spectrum of the Floquet evolution operator $\mathbf{U}_\mathrm{F}(T)$ with $\eta = \mathrm{i}/2$ (not at root of unity) and system size $L = 6$. $T = \frac{\pi + 2\mathrm{i}\eta - 0.1}{2\cosh\eta}$, $\frac{\pi + 2\mathrm{i}\eta}{2\cosh\eta}$ (phase transition), $\frac{\pi + 2\mathrm{i}\eta + 0.1}{2\cosh\eta}$ for the panels from left to right respectively.

In the case $\eta \in (0, \frac{\pi}{2}]\mathrm{i}$, we define two separate regimes

$$\mathrm{I}: \ \alpha \in \mathbb{R} \ \Rightarrow \ T \in \left[0, \frac{\pi + 2\mathrm{i}\eta}{2\cosh\eta}\right] \cup \left[\frac{\pi - 2\mathrm{i}\eta}{2\cosh\eta}, \frac{\pi}{\cosh\eta}\right), \tag{5.53}$$

and

$$\mathrm{II}: \ \mathrm{Im}\,\alpha = \frac{\pi}{2} \ \Rightarrow \ T \in \left(\frac{\pi + 2\mathrm{i}\eta}{2\cosh\eta}, \frac{\pi - 2\mathrm{i}\eta}{2\cosh\eta}\right), \tag{5.54}$$

where the period $T$ is defined modulo $\frac{\pi}{\cosh\eta}$. In Regime I we have $\mathrm{Im}\,\alpha = 0$, while in Regime II one has $\mathrm{Im}\,\alpha = \frac{\pi}{2}$.

Similarly, for $\eta \in (\frac{\pi}{2}, \pi)\mathrm{i}$, we define the two regimes accordingly

$$\mathrm{I}: \ \alpha \in \mathbb{R} \ \Rightarrow \ T \in \left[0, \frac{-\pi - 2\mathrm{i}\eta}{2\cosh\eta}\right] \cup \left[\frac{3\pi + 2\mathrm{i}\eta}{2\cosh\eta}, \frac{\pi}{\cosh\eta}\right), \tag{5.55}$$

and

$$\mathrm{II}: \ \mathrm{Im}\,\alpha = \frac{\pi}{2} \ \Rightarrow \ T \in \left(\frac{-\pi - 2\mathrm{i}\eta}{2\cosh\eta}, \frac{3\pi + 2\mathrm{i}\eta}{2\cosh\eta}\right). \tag{5.56}$$

From (5.48), we know that in Regime I the eigenvalues of the Floquet evolution operator are uni-modular (the eigenvalues of the Floquet Hamiltonian are real). However, we cannot generalise the argument to the Regime II. Instead, we numerically diagonalise the Floquet evolution operators with *finite system sizes*, as described in Figs. 10 and 12. Interestingly, we find out that the eigenvalues of the Floquet evolution operator are no longer uni-modular in Regime II. When taking into account the Floquet Hamiltonian, the results above are equivalent to the fact that the eigenvalues of the Floquet Hamiltonian are complex in Regime II. Moreover, the same behaviour occurs with any system size $L \bmod 2 = 0$. We shall expand on this point in Sec. 6.2.

# 6 Dynamical anti-unitary symmetry breaking in the easy-plane regime

In this section we concentrate on the easy-plane regime, i.e. $\eta \in \mathrm{i}\mathbb{R}/\{0\}$ ($|\Delta| < 1$).

## 6.1 Anti-unitary symmetry of the Floquet Hamiltonian

As we have shown in the previous sections, the Floquet Hamiltonian $\mathbf{H}_\mathrm{F}$ is not unitary when we are in the easy-plane regime. In addition, we show that the Floquet Hamiltonian

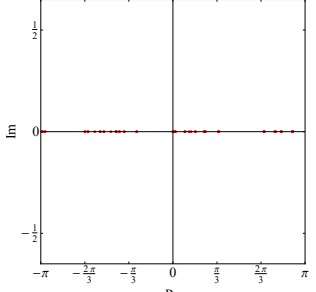 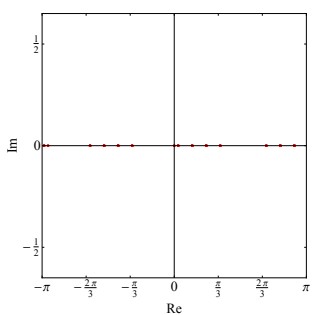 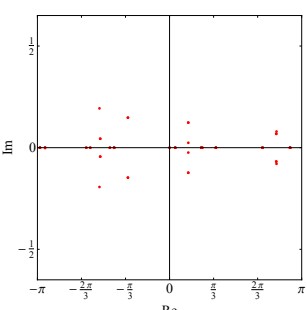

Figure 11: Spectra of the Floquet Hamiltonian $\mathbf{H}_F(T)$ with $\eta = i/2$ (not at root of unity) and system size $L = 6$. $T = \frac{\pi + 2i\eta - 0.1}{2\cosh\eta}$, $\frac{\pi + 2i\eta}{2\cosh\eta}$ (phase transition), $\frac{\pi + 2i\eta + 0.1}{2\cosh\eta}$ for the figures from left to right respectively.

has a real spectrum in Regime I of Fig. 14. There have been numerous studies on the non-Hermitian systems with real spectra, which usually possess the so-called $\mathcal{PT}$ symmetry (or more generally anti-unitary symmetry) [107–110]. In the case of the Floquet Hamiltonian that we consider, it commutes with an anti-unitary operator, which can be seen as a generalisation of the usual $\mathcal{PT}$ symmetry. We demonstrate the anti-unitary symmetry of the Floquet Hamiltonian $\mathbf{H}_F$ as follows.

To begin with, we define the parity and time reversal symmetries on quantum lattice models. The parity symmetry is defined as

$$\mathcal{P}: \quad \mathbf{O}_m \to \mathbf{O}_{L-m+1}, \tag{6.1}$$

or equivalently, the parity conjugation operator is

$$\mathcal{P} = \prod_{m=1}^{L/2} \mathbf{P}_{m,L-m+1}. \tag{6.2}$$

From its definition, parity conjugation operator is unitary.

As for the time reversal symmetry, it is defined as

$$\mathcal{T}: \quad \mathbf{O}_m \to \bar{\mathbf{O}}_m, \tag{6.3}$$

which is anti-unitary [111, 112].

Using the properties of two symmetries, we arrive at

$$\mathcal{P}^2 = \mathcal{T}^2 = \mathbb{1}, \quad [\mathcal{P}, \mathcal{T}] = 0. \tag{6.4}$$

We define an anti-unitary operator $\mathcal{A}$ as the combination of the $\mathcal{PT}$ operator and the right translation opeartor $\mathbf{G}$ (5.9),

$$\mathcal{A} = \mathbf{G}\mathcal{PT}, \quad \mathcal{A}^{-1} = \mathcal{PT}\mathbf{G}^{-1}. \tag{6.5}$$

In the following, we shall show that the (non-Hermitian) Floquet Hamiltonian in both Regime I and II commutes with the anti-unitary operator $\mathcal{A}$.

From the properties of the aTL generators (5.1), we have

$$\mathcal{P}\mathbf{e}_{m,m+1}\mathcal{P} = \mathbf{e}_{L-m,L-m+1}; \tag{6.6}$$

$$\mathcal{T}\mathbf{e}_{a,b}\mathcal{T} = \bar{\mathbf{e}}_{a,b} = \frac{q + q^{-1}}{2} - \left(\sigma_a^x\sigma_b^x + \sigma_a^y\sigma_b^y + \frac{q + q^{-1}}{2}\sigma_a^z\sigma_b^z\right) - \frac{q - q^{-1}}{2}(\sigma_b^z - \sigma_a^z)$$

$$= \mathbf{e}_{b,a}, \quad \eta \in i\mathbb{R}. \tag{6.7}$$

As for the inhomogeneity $\alpha \in \mathbb{C}$, we have

$$\mathcal{T}\frac{\sinh \alpha}{\sinh(\eta - \alpha)}\mathcal{T} = -\frac{\sinh \bar{\alpha}}{\sinh(\eta + \bar{\alpha})}, \quad \forall \eta \in i\mathbb{R}/\{0\}. \tag{6.8}$$

Hence, acting the operator $\mathcal{A}$ to the R matrix, we obtain

$$\mathcal{A}\check{\mathbf{R}}_{m,m+1}(-\alpha)\mathcal{A}^{-1} = \check{\mathbf{R}}_{L-m+1,L-m+2}(\bar{\alpha}). \tag{6.9}$$

Now we would like to investigate how the Floquet evolution operator changes under the action of anti-unitary operator $\mathcal{A}$. From (4.1), we decompose the Floquet evolution operator into two parts,

$$\mathbf{U}_{\mathrm{F}}(T) = \mathbf{U}_{\mathrm{F}}(\{0, \alpha\}) = \mathbf{V}_2(\alpha)\mathbf{V}_1(\alpha),$$

$$\mathbf{V}_2(\alpha) = \prod_{m=1}^{L/2} \check{\mathbf{R}}_{2m-1,2m}(-\alpha), \quad \mathbf{V}_1(\alpha) = \prod_{m=1}^{L/2} \check{\mathbf{R}}_{2m,2m+1}(-\alpha). \tag{6.10}$$

We start with $\mathbf{V}_2(\alpha)$, i.e.

$$\mathcal{A}\mathbf{V}_2(\alpha)\mathcal{A}^{-1} = \mathcal{A}\prod_{m=1}^{L/2} \check{\mathbf{R}}_{2m-1,2m}(-\bar{\alpha})\mathcal{A}^{-1}$$

$$= \prod_{m=1}^{L/2} \check{\mathbf{R}}_{L-2m,L-2m+1}(\bar{\alpha}) = \mathbf{V}_1(-\bar{\alpha}). \tag{6.11}$$

Similarly, for the operator $\mathbf{V}_1(\alpha)$,

$$\mathcal{A}\mathbf{V}_1(\alpha)\mathcal{A}^{-1} = \mathbf{V}_2(-\bar{\alpha}). \tag{6.12}$$

Together we obtain

$$\mathcal{A}\mathbf{U}_{\mathrm{F}}(\alpha)\mathcal{A}^{-1} = \mathbf{V}_1(-\bar{\alpha})\mathbf{V}_2(-\bar{\alpha}). \tag{6.13}$$

Moreover, it is straightforward to derive the following inversion formulae

$$\mathbf{V}_j(\alpha)\mathbf{V}_j(-\alpha) = \mathbf{V}_j(-\alpha)\mathbf{V}_j(\alpha) = \mathbb{1}, \quad j \in \{1, 2\}, \tag{6.14}$$

from (5.42).

If we consider the case when $\alpha \in \mathbb{R}$, we obtain

$$\mathcal{A}\mathbf{U}_{\mathrm{F}}(\alpha)\mathcal{A}^{-1} = \mathcal{A}\mathbf{V}(\alpha)\mathbf{W}(\alpha)\mathcal{A}^{-1} = \mathbf{W}(-\alpha)\mathbf{V}(-\alpha) = \mathbf{U}_{\mathrm{F}}^{-1}(\alpha). \tag{6.15}$$

Since the Floquet evolution operator is the exponential of the Floquet Hamiltonian, we have

$$\mathcal{A}\mathbf{U}_{\mathrm{F}}(T)\mathcal{A}^{-1} = \mathcal{A}\exp\left(-2i\mathbf{H}_{\mathrm{F}}(T)T\right)\mathcal{A}^{-1}$$

$$= \exp\left(2i\mathcal{A}\mathbf{H}_{\mathrm{F}}(T)\mathcal{A}^{-1}T\right)$$

$$= \exp\left(2i\mathbf{H}_{\mathrm{F}}(T)T\right) = \mathbf{U}_{\mathrm{F}}^{-1}(T). \tag{6.16}$$

This implies that the Floquet Hamiltonian is invariant under the anti-unitary operator $\mathcal{A}$ when $\alpha \in \mathbb{R}$, i.e.

$$\mathcal{A}\mathbf{H}_{\mathrm{F}}(T)\mathcal{A}^{-1} = \mathbf{H}_{\mathrm{F}}(T), \tag{6.17}$$

when $\alpha \in \mathbb{R}$.

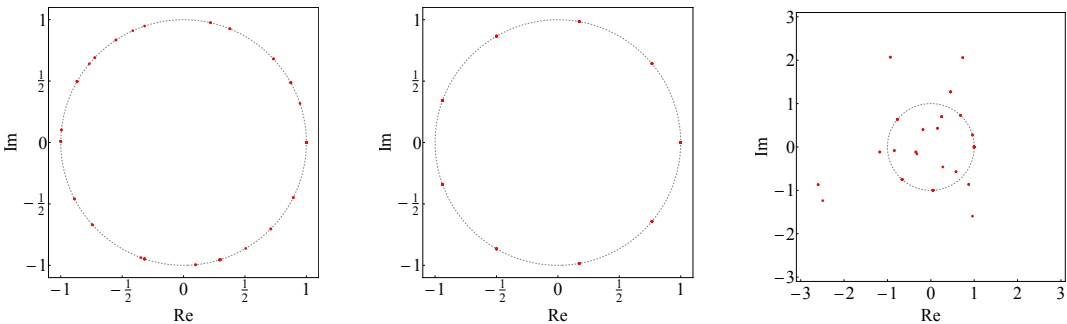

Figure 12: Spectra of the Floquet evolution operator $\mathbf{U}_{\mathrm{F}}(T)$ with $\eta = \mathrm{i}\pi/3$ (at root of unity) and system size $L = 6$. $T = \frac{\pi + 2\mathrm{i}\eta - 0.1}{2\cosh\eta}$, $\frac{\pi + 2\mathrm{i}\eta}{2\cosh\eta}$ (phase transition), $\frac{\pi + 2\mathrm{i}\eta + 0.1}{2\cosh\eta}$ for the figures from left to right respectively.

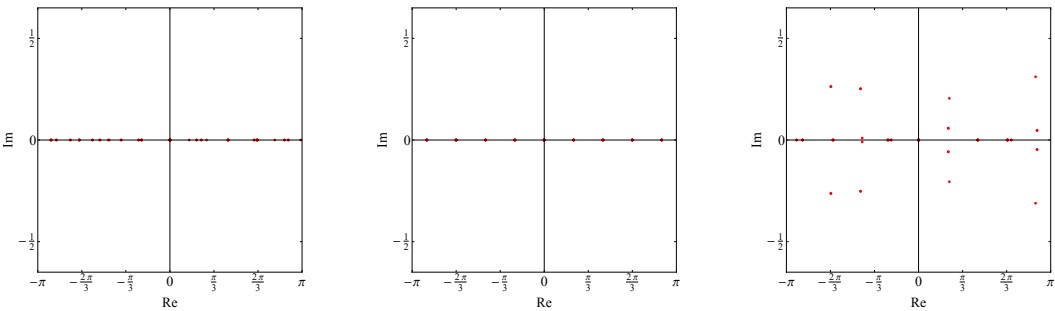

Figure 13: Spectra of the Floquet Hamiltonian $\mathbf{H}_{\mathrm{F}}(T)$ with $\eta = \mathrm{i}\pi/3$ (at root of unity) and system size $L = 6$. $T = \frac{\pi + 2\mathrm{i}\eta - 0.1}{2\cosh\eta}$, $\frac{\pi + 2\mathrm{i}\eta}{2\cosh\eta}$ (phase transition), $\frac{\pi + 2\mathrm{i}\eta + 0.1}{2\cosh\eta}$ for the figures from left to right respectively.

Moreover, when $\operatorname{Im}\alpha = \frac{\pi}{2}$, we have

$$\bar{\alpha} = \alpha - \mathrm{i}\pi. \tag{6.18}$$

Since $\alpha$ is defined up to $\mathrm{i}\pi$, cf. (5.38), we have

$$\mathbf{U}_{\mathrm{F}}(\{0, -\bar{\alpha}\}) = \mathbf{U}_{\mathrm{F}}(\{0, -\alpha\}), \quad \alpha - \mathrm{i}\frac{\pi}{2} \in \mathbb{R}, \tag{6.19}$$

i.e.

$$\begin{aligned}
\mathcal{A}\mathbf{U}_{\mathrm{F}}(T)\mathcal{A}^{-1} &= \mathbf{U}_{\mathrm{F}}^{-1}(T), \\
\mathcal{A}\mathbf{H}_{\mathrm{F}}(T)\mathcal{A}^{-1} &= \mathbf{H}_{\mathrm{F}}(T), \quad \alpha - \mathrm{i}\frac{\pi}{2} \in \mathbb{R}.
\end{aligned} \tag{6.20}$$

Since in Regime I $\alpha \in \mathbb{R}$ and in Regime II $\operatorname{Im}\alpha = \frac{\pi}{2}$, the Floquet Hamiltonian $\mathbf{H}_{\mathrm{F}}$ commutes with $\mathcal{A}$ with all possible values of the Floquet period $2T \in \mathbb{R}_+$ in the easy-plane regime, despite being non-Hermitian.

## 6.2 Dynamical breaking of anti-unitary symmetry in the easy-plane regime

Even though the Floquet Hamiltonian $\mathbf{H}_{\mathrm{F}}$ commutes with the anti-unitary operator $\mathcal{A}$ as shown above, it is not enough to assert that all the (right) eigenstates of the Floquet Hamiltonian are also the eigenstates of the anti-unitary operator $\mathcal{A}$ [113].

However, we can already deduce some properties of the eigenvalues of the Floquet Hamiltonian $\mathbf{H}_{\mathrm{F}}$. Assume that all the (right) eigenstates $|\psi\rangle$ of the Floquet Hamiltonian

$\mathbf{H}_\mathrm{F}(T)$ are also eigenstates of the anti-unitary operator $\mathcal{A}$ in certain range of values of $T$, i.e.

$$\mathbf{H}_\mathrm{F}(T)|\psi\rangle = E_\psi|\psi\rangle, \quad \mathcal{A}|\psi\rangle = \lambda|\psi\rangle. \tag{6.21}$$

Therefore, all the eigenvalues in this case must be real, i.e.

$$\mathcal{A}\mathbf{H}_\mathrm{F}(T)|\psi\rangle = \bar{E}_\psi \mathcal{A}|\psi\rangle = E_\psi \mathcal{A}|\psi\rangle, \tag{6.22}$$

i.e. $E_\psi = \bar{E}_\psi \in \mathbb{R}$. We denote this phase as the $\mathbf{G}\mathcal{PT}$ symmetric phase, since the anti-unitary symmetry is preserved (not broken) for all the eigenstates.

However, if there exist eigenstates of the Floquet Hamiltonian that are not eigenstates of the anti-unitary operator $\mathcal{A}$ in another phase, we denote the phase as the $\mathbf{G}\mathcal{PT}$ symmetry *broken* phase. In this scenario, we start with an eigenstate of the Floquet Hamiltonian $|\psi\rangle$ which is not an eigenstate of $\mathcal{A}$,

$$\mathbf{H}_\mathrm{F}(T)|\psi\rangle = E_\psi|\psi\rangle, \quad \mathcal{A}|\psi\rangle = |\varphi\rangle \varpropto |\psi\rangle. \tag{6.23}$$

In this case, we have

$$\mathbf{H}_\mathrm{F}(T)|\varphi\rangle = \mathcal{A}\mathbf{H}_\mathrm{F}(T)|\psi\rangle = \bar{E}_\psi|\varphi\rangle, \tag{6.24}$$

i.e. the spectrum of $\mathbf{H}_\mathrm{F}(T)$ consists of complex conjugate pairs in the $\mathbf{G}\mathcal{PT}$ symmetry broken phase.

When the condition (5.53) or (5.55) is satisfied, i.e. in Regime I, the spectrum of $\mathbf{U}_\mathrm{F}(T)$ is uni-modular and the spectrum of the Floquet Hamiltonian $\mathbf{H}_\mathrm{F}(T)$ is real, as demonstrated in the left figures of Figs. 10, 11 and 12, 13. Therefore, Regime I corresponds to the $\mathbf{G}\mathcal{PT}$ symmetric phase. Meanwhile, when the condition (5.54) or (5.56) is satisfied, i.e. in Regime II, due to the anti-unitary symmetry, the spectrum of the Floquet Hamiltonian $\mathbf{H}_\mathrm{F}(T)$ is no longer real, but consists of complex conjugate pairs, as shown in the right figures of Figs. 11 and 13. Hence, Regime II corresponds to the $\mathbf{G}\mathcal{PT}$ broken phase. We expect a "*phase transition*" occurring between the two phases. This phenomenon is demonstrated in Figs. 10, 11 and 12, 13: in Regime I the spectrum of $\mathbf{U}_\mathrm{F}(T)$ is uni-modular, denoting the $\mathbf{G}\mathcal{PT}$ symmetric phase, cf. the left figures of Figs. 10 and 12; when $T = \frac{\pi \pm 2\mathrm{i}\eta}{2\cosh\eta}$ (mod $\frac{\pi}{\cosh\eta}$), i.e. the phase transition, the eigenvalues of $\mathbf{U}_\mathrm{F}(T)$ attain additional discrete symmetries, cf. the middle figures of Figs. 10 and 12; in Regime II the spectrum of $\mathbf{U}_\mathrm{F}(T)$ is no longer uni-modular, denoting the $\mathbf{G}\mathcal{PT}$ broken phase, cf. the right figures of Figs. 10 and 12.

**Remarks.** The phase transition that we observe here is different from the conventional phase transition originated from the spontaneous breaking of a unitary symmetry in a Hermitian system. First of all, the phase transition here is a property of the entire spectrum, while the conventional phase transition is usually about the property of the ground state(s). Furthermore, the phase transition here does not require the thermodynamic limit (i.e. system size goes to infinity), as we observe the same behaviour of the spectra of the Floquet evolution operator for different system sizes. Moreover, at root of unity $\eta = \mathrm{i}\pi/2$ ($\beta = 0$), we need to analyses the phases differently, since (5.6) does not apply any more. In that case, the phase transition is still present, but with finite $T$ ($\beta T \to 0$), which is discussed in [41].

## 6.3 Phase transition at root of unity: conjecture of the spectra

As we discussed in the previous sections, the phase transition between the anti-unitary symmetry preserved and broken phases happens throughout the entire easy-plane regime.

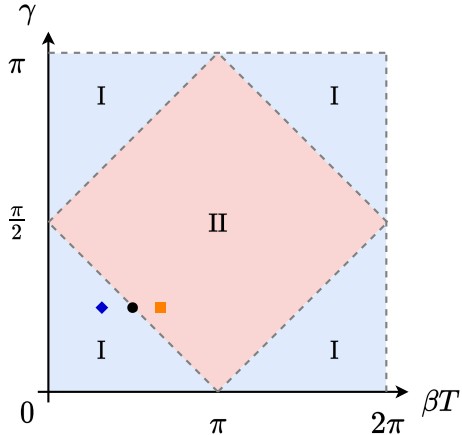
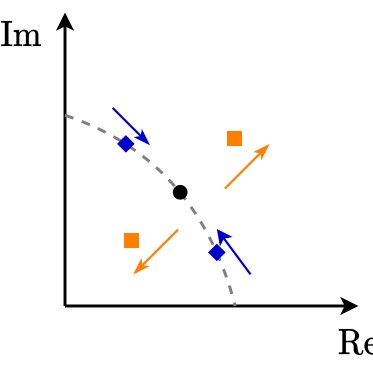

Figure 14: Left figure: Phase diagram in the easy-plane regime. When the system is in Regime I, the spectrum of the Floquet Hamiltonian is real and it possesses the $\mathbf{G}\mathcal{PT}$ symmetry. On the contrary, in Regime II, the spectrum of the Floquet Hamiltonian is complex and the $\mathbf{G}\mathcal{PT}$ symmetry is broken. Anisotropic parameter $\eta = \mathrm{i}\gamma$. Right figure: Demonstration of how the eigenvalues of $\mathbf{U}_{\mathrm{F}}(T)$ change across the phase transition. We start with eigenvalues of $\mathbf{U}_{\mathrm{F}}$ within Regime I, which are depicted as blue diamonds. When we are approaching the phase transition, the blue diamonds move toward each other and becomes degenerate at the black dot at phase transition (which is the root of unity value conjectured in (6.27) if the anisotropy is at root of unity too). When we cross the phase transition into Regime II, the eigenvalues (as the orange squares) are no longer necessarily uni-modular and they move away from the black dot in the complex plane.

In addition, we observe that the spectra of the Floquet evolution operators behave in a special way at root of unity anisotropies, i.e.

$$q^{\varepsilon \ell_2} = 1, \quad \eta = \mathrm{i}\pi\frac{\ell_1}{\ell_2}, \quad \text{with } \varepsilon = \begin{cases} 2 & \text{if } \ell_1 \text{ is odd,} \\ 1 & \text{if } \ell_1 \text{ is even.} \end{cases} \tag{6.25}$$

More specifically, when we are at the phase transition point, i.e. $\alpha \to \pm\infty$, the Floquet evolution operator becomes

$$\mathbf{U}_{\mathrm{F}}(\{0, \pm\infty\}) = \prod_{m=1}^{L/2} \left(\mathbb{1} - e^{\mp\eta}\mathbf{e}_{2m-1,2m}\right) \prod_{m=1}^{L/2} \left(\mathbb{1} - e^{\mp\eta}\mathbf{e}_{2m,2m+1}\right). \tag{6.26}$$

The eigenvalues of this scenario are shown in the middle figures of Figs. 10 and 12.

Interestingly, when the anisotropy $\Delta$ is at root of unity value, i.e. the case of Fig. 12, we conjecture that eigenvalues of $\mathbf{U}_{\mathrm{F}}$ locates at a different set of roots of unity depending on both the denominator $\ell_2$ and the system size $L$, i.e.

$$\exp\left(\frac{2\mathrm{i}\pi n}{\ell_2 L/2}\right), \quad n \in \mathbb{Z}. \tag{6.27}$$

**Remark.** The conjecture that the eigenvalues of $\mathbf{U}_{\mathrm{F}}$ at phase transition locate at the root of unity values is further exemplified in the second figure of Fig. 12, where the eigenvalues are equally distributed at $\exp\left(\frac{2\mathrm{i}\pi n}{9}\right)$, where $9 = \ell_2 \times \frac{L}{2}$. However, in the general case, e.g. Fig. 15, not all possible root of unity values appear in the spectra of $\mathbf{U}_{\mathrm{F}}(\pm\infty)$, while the degeneracies at each eigenvalues might vary too.

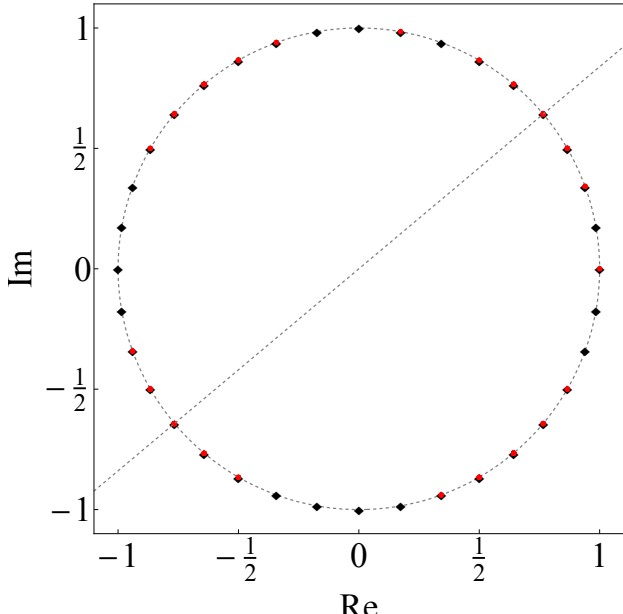

Figure 15: Spectra with $L = 8$, $\eta = \frac{5i\pi}{9}$. Black diamonds are all the possible root of unity values of $\exp\left(\frac{2i\pi n}{\ell_2 L/2}\right)$, $n \in \mathbb{Z}$. Red dots are the ones that are present in the spectra of $\mathbf{U}_{\mathrm{F}}(+\infty)$. The spectrum is mirror symmetric, revealing further symmetries hidden in the spectrum.

We can infer part of the conjecture from the relation between $\mathbf{U}_{\mathrm{F}}$ and the staggered transfer matrix. Since

$$\mathbf{U}_{\mathrm{F}}(\{0, \pm\infty\}) = \lim_{\alpha \to \infty} \frac{1}{\sinh^L \eta \sinh^L(\eta - \alpha)} \mathbf{T}^2(0, \alpha) \mathbf{G}^2, \tag{6.28}$$

the eigenvalues of $\mathbf{U}_{\mathrm{F}}(\pm\infty)$ are proportional to the ones of $\mathbf{G}^2 = \exp(i\mathbf{p})$, i.e. the exponential of the total momentum. The eigenvalues of $\mathbf{G}^2$ are

$$\exp\left(\frac{2i\pi n}{L/2}\right), \tag{6.29}$$

which consists of a part of the conjectural values (6.27). The eigenvalues of the part that proportional to $\mathbf{T}^2(0, \alpha)$ in principle can be obtained from (5.38) by taking the limit $\alpha \to \infty$. However, we cannot naively take for granted that Bethe roots $u_j$ are of order 1. In fact, the existence of Bethe roots located at infinity makes it nontrivial taking the limit $\alpha \to \pm\infty$. We postpone the proof of the conjecture (6.27) and the study of the spectra of the Floquet evolution operators at phase transitions to future investigations.

Instead, we would like to offer a phenomenological description of the phase transitions. The eigenvalues of the Floquet evolution operator $\mathbf{U}_{\mathrm{F}}$ move toward the eigenvalues at the phase transition as the Floquet period $T$ approaches the phase transition point from Regime I. At the phase transition, the eigenvalues of $\mathbf{U}_{\mathrm{F}}$ become degenerate. Once the Floquet period $T$ moves away from the phase transition and is inside Regime II, the eigenvalues repel and move away from the unit circle, corresponding to the dynamical breaking of the anti-unitary symmetry. This procedure is illustrated in the right figure of Fig. 14. Beware that the right figure of Fig. 14 is meant as only an illustration of the phenomenon. In fact, there might be more than two eigenvalues become degenerate at the phase transition. And there might be other additional degeneracies within both phases that are not due to the

mechanism above. At the phase transition, the spectrum of $\mathbf{U}_F$ obtains extra degeneracies, corresponding to the appearance of the Jordan blocks in the Floquet Hamiltonian. Further discussions will be postponed to later work.

# 7  Conclusion

In this paper, we have proven a generic method of constructing the integrable Floquet circuits with depth $n \geq 2$ from the inhomogeneous transfer matrices using the Floquet Baxterisation. Our proof does not require any specific properties for the R matrices, for instance the regularity condition and the difference form of the spectral parameter. Our proof generalises several known cases with the depth $n = 2$. When the depth $n \geq 3$, the Floquet Baxterisation provides a systematic way of obtaining the integrable Floquet evolution operators that has not been discussed before. Moreover, one can interpret the integrable Floquet evolution operator as a non-reciprocal tilted transfer matrix of integrable lattice statistical-mechanical models. In addition to the periodic case, we also prove a similar method of constructing integrable Floquet dynamics with open boundary condition using the boundary Yang–Baxter relations.

After proving the general methods of constructing integrable Floquet dynamics, we focus on the explicit example of the Floquet Baxterisation using the R matrix of the 6-vertex model, which confirms parts of the conjecture on the Floquet integrability from the (affine) Temperley–Lieb algebra in [26] motivated from the explicit calculations on the local density of conserved charges.

When the 6-vertex model is in the easy-axis and isotropic regimes, the Floquet evolution operator is unitary, alas the Floquet Hamiltonian is Hermitian, leading to a unitary discrete time evolution. However, when we concentrate on the easy-plane regime, the Floquet evolution operator is no longer unitary (hence the Floquet Hamiltonian is non-Hermitian). The Floquet Hamiltonian in this case possesses an anti-unitary symmetry. The anti-unitary symmetry can be broken with respect to different Floquet period $T$, leading to a diamond-shape phase diagram shown in Fig. 14. In Regime I of Fig. 14, the anti-unitary symmetry is not broken, and the Floquet Hamiltonian is pseudo-Hermitian with a real spectrum, leading to a norm preserving time evolution. On the other hand, in Regime II of Fig. 14, the anti-unitary symmetry is broken, resulting in a Floquet Hamiltonian with a spectrum consisting of complex conjugate pairs. The phase transition between two regimes happens even with finite system sizes, with interesting behaviours at root of unity values of anisotropy.

Equipped with the results obtained in this paper, we are ready to focus on other intriguing questions. For instance, we can use the Bethe ansatz technique to further investigate the quantum quenches of certain initial states with the integrable Floquet circuits, which are expected to have different appearances for physical quantities such as the correlation functions and the entanglement entropies. It seems promising for developing the hydrodynamic theory for the integrable Floquet circuits, since the existence of the extensively many local (and quasi-local) conserved quantities are accessible from the underlying inhomogeneous transfer matrices.

Furthermore, the integrable Floquet circuits with depth $n \geq 3$ have not been systematically studied. From the procedure of the Floquet Baxterisation, we offer a new perspective, different from the existing constructions of the quantum cellular automaton. It would be useful to understand the similarities and differences of these constructions. Moreover, we would like to understand better the semi-classical limit of the integrable Floquet circuits considered in this paper, which should correspond to the discrete space-time classical

integrable models studied in [114, 115] when depth $n = 2$. With numerous open questions, we conclude that the Floquet Baxterisation offers us extraordinary opportunities to understand better many aspects of out-of-equilibrium physics and exactly solvable models.

## Acknowledgments

Y.M. thanks Kemal Bidzhiev, Jules Lamers, Vincent Pasquier and Lenart Zadnik for fruitful discussions. Y.M. is grateful to Philippe Di Francesco for suggesting the generalisation of the Floquet Baxterisation protocol to arbitrary depths. Part of the work has been conducted during the workshop "Randomness, Integrability, and Universality" at GGI. Y.M. acknowledges the support from the GGI BOOST fellowship. The work by V.G. is part of the DeltaITP consortium, a program of the Netherlands Organization for Scientific Research (NWO) funded by the Dutch Ministry of Education, Culture and Science (OCW). This study is also supported by the Russian Science Foundation (Grant No. 20-42-05002, work of D.V.K.).

# A   New solution to the set-theoretical Yang–Baxter equation

In this Appendix we present the solutions to the set-theoretical Yang–Baxter equation using the substitution rules (3.12).

Using the substitution rule (3.12), we consider the triple $(x_1, x_2, x_3)$. From the left-hand side of (3.11) we have

$$R_{23}(x_1, x_2, x_3) = (x_1, x_2^n x_3^m, x_2^p x_3^q), \tag{A.1}$$

$$
\begin{aligned}
R_{13}(R_{23}(x_1, x_2, x_3)) &= R_{12}(x_1, x_2^n x_3^m, x_2^p x_3^q) \\
&= (x_1^n (x_2^p x_3^q)^m, x_2^n x_3^m, x_1^p (x_2^p x_3^q)^q) \\
&= (x_1^n x_2^{pm} x_3^{qm}, x_2^n x_3^m, x_1^p x_2^{pq} x_3^{q^2}),
\end{aligned}
\tag{A.2}
$$

$$
\begin{aligned}
R_{12}(R_{13}(R_{23}(x_1, x_2, x_3))) &= R_{12}(x_1^n x_2^{pm} x_3^{qm}, x_2^n x_3^m, x_1^p x_2^{pq} x_3^{q^2}) \\
&= ((x_1^n x_2^{pm} x_3^{qm})^n (x_2^n x_3^m)^m, (x_1^n x_2^{pm} x_3^{qm})^p (x_2^n x_3^m)^q, x_1^p x_2^{pq} x_3^{q^2}) \\
&= (x_1^{n^2} x_2^{pnm+nm} x_3^{qmn+m^2}, x_1^{np} x_2^{p^2 m+nq} x_3^{qmp+mq}, x_1^p x_2^{pq} x_3^{q^2}).
\end{aligned}
\tag{A.3}
$$

Similarly, from the right-hand side of (3.11) we obtain

$$R_{12}(x_1, x_2, x_3) = (x_1^n x_2^m, x_1^p x_2^q, x_3), \tag{A.4}$$

$$
\begin{aligned}
R_{13}(R_{12}(x_1, x_2, x_3)) &= R_{13}(x_1^n x_2^m, x_1^p x_2^q, x_3) \\
&= ((x_1^n x_2^m)^n x_3^m, x_1^p x_2^q, (x_1^n x_2^m)^p x_3^q) \\
&= (x_1^{n^2} x_2^{mn} x_3^m, x_1^p x_2^q, x_1^{np} x_2^{mp} x_3^q),
\end{aligned}
\tag{A.5}
$$

$$
\begin{aligned}
R_{23}(R_{13}(R_{12}(x_1, x_2, x_3))) &= R_{23}((x_1^{n^2} x_2^{mn} x_3^m, x_1^p x_2^q, x_1^{np} x_2^{mp} x_3^q) \\
&= (x_1^{n^2} x_2^{mn} x_3^m, (x_1^p x_2^q)^n (x_1^{np} x_2^{mp} x_3^q)^m, (x_1^p x_2^q)^p (x_1^{np} x_2^{mp} x_3^q)^q) \\
&= (x_1^{n^2} x_2^{mn} x_3^m, x_1^{pn+npm} x_2^{qn+m^2 p} x_3^{qm}, x_1^{p^2 npq} x_2^{qp+mqp} x_3^{q^2}).
\end{aligned}
\tag{A.6}
$$

Comparing both sides, we arrive at the conclusion that the following requirements have to be satisfied

$$n^2 = n^2, \quad pn + pnm = np, \quad p^2 + npq = p,$$
$$mn = pmn + mn, \quad qn + m^2 p = p^2 m + nq, \quad qp + mqp = qp, \qquad \text{(A.7)}$$
$$m = qmn + m^2, \quad qm = qmp + qm, \quad q^2 = q^2,$$

in order to guarantee that the set-theoretical Yang–Baxter equation (3.11) is fulfilled. Note that there are $n \leftrightarrow q$ and $p \leftrightarrow m$ symmetries in these equations.

This remark and a straightforward analysis leads to the following classes of solutions [where we use the notation $(n, m, p, q)$]:

$$
\begin{aligned}
A_{nq} &: (n, 0, 0, q), \\
B_f &: (n, 1 - nq, 0, q), \\
B_g &: (n, 0, 1 - nq, q), \\
C_q &: (0, 0, 1, q), \\
D_q &: (0, 1, 0, q), \\
C_n &: (n, 0, 1, 0), \\
D_n &: (n, 1, 0, 0).
\end{aligned}
\qquad \text{(A.8)}
$$

In fact, the equations above can be further reduced to the following four classes with arbitrary parameters $n$ and $q$, i.e.

$$
\begin{aligned}
A_{nq} &: (n, 0, 0, q), \\
B_f &: (n, 1 - nq, 0, q), \\
B_g &: (n, 0, 1 - nq, q) \\
P &: (0, 1, 1, 0).
\end{aligned}
\qquad \text{(A.9)}
$$

# B  The allowed value for Bethe roots in the easy-plane regime

From the definition of the staggered transfer matrix (5.12), we obtain the following identity in the easy-plane regime $\eta \in i\mathbb{R}/\{0\}$, i.e.

$$\mathbf{T}^\dagger(u, \alpha, \eta) = \prod_{m=1}^{L} \sigma_m^x \mathbf{T}(\bar{u} - \eta, \bar{\alpha}, \eta) \prod_{m=1}^{L} \sigma_m^x, \qquad \text{(B.1)}$$

We consider the case with $\alpha = \bar{\alpha} \in \mathbb{R}$. In that case, the eigenvalues of $\mathbf{T}^\dagger(u, \alpha, \eta)$ and $\mathbf{T}(\bar{u} - \eta, \alpha, \eta)$ coincide. (though the eigenvectors are related by the spin-flip operator.) Therefore, we expect the eigenvalues of the staggered transfer matrix

$$\bar{\tau}(u, \alpha, \{u_m\}_{m=1}^{M}) = \tau(\bar{u} - \eta, \alpha, \{u_m\}_{m=1}^{M}), \qquad \text{(B.2)}$$

which in terms of the spectral parameter $\lambda$ becomes

$$\bar{\tau}(\lambda, \alpha, \{\lambda_m\}_{m=1}^{M}) = \tau(\bar{\lambda}, \alpha, \{\lambda_m\}_{m=1}^{M}). \qquad \text{(B.3)}$$

We consider the case with $\lambda = \bar{\lambda}$, which leads us to the following equation,

$$\left[\sinh\left(\lambda + \frac{\alpha}{2} + \frac{\eta}{2}\right)\sinh\left(\lambda - \frac{\alpha}{2} + \frac{\eta}{2}\right)\right]^{L/2}\left(\prod_{m=1}^{M}\sinh(\lambda - \lambda_m - \eta)\sinh\left(\lambda - \bar{\lambda}_m\right)-\right.$$

$$\left.\prod_{m=1}^{M}\sinh\left(\lambda - \bar{\lambda}_m - \eta\right)\sinh(\lambda - \lambda_m)\right) =$$

$$\left[\sinh\left(\lambda + \frac{\alpha}{2} - \frac{\eta}{2}\right)\sinh\left(\lambda - \frac{\alpha}{2} - \frac{\eta}{2}\right)\right]^{L/2}\left(\prod_{m=1}^{M}\sinh\left(\lambda - \bar{\lambda}_m + \eta\right)\sinh(\lambda - \lambda_m)-\right.$$

$$\left.\prod_{m=1}^{M}\sinh(\lambda - \lambda_m + \eta)\sinh\left(\lambda - \bar{\lambda}_m\right)\right). \tag{B.4}$$

Comparing the zeros of the trigonometric polynomials on both sides of the equation, we realise that

$$\prod_{m=1}^{M}\sinh(\lambda - \lambda_m - \eta)\sinh\left(\lambda - \bar{\lambda}_m\right) = \prod_{m=1}^{M}\sinh\left(\lambda - \bar{\lambda}_m - \eta\right)\sinh(\lambda - \lambda_m). \tag{B.5}$$

In order to matching the zeros of the trigonometric polynomials on each side of the equation, we can conclude that the Bethe roots $\lambda_n$ satisfies one of the three conditions

- $\lambda_n \in \mathbb{R}$,

- $\text{Im } \lambda_n = \frac{\pi}{2}$,

- If $\lambda_n \in \{\lambda_m\}_{m=1}^{M}$, $\bar{\lambda}_n \in \{\lambda_m\}_{m=1}^{M}$ .

## C  Isotropic limit

When we consider the isotropic case ($|\Delta| = 1$), the R matrix consists of rational functions instead of trigonometric functions in terms of the spectral parameter. Here we only focus on the case with $\Delta = 1$ ($\eta = 0$). The case with $\Delta = -1$ ($\eta = i\pi$) can be obtained analogously. More specifically, the R matrix becomes

$$\check{\mathbf{R}}_{a,b}^{\text{iso}}(u) = \mathbb{1} - \frac{u}{u + i}\mathbf{e}_{a,b}, \quad \mathbf{e}_{a,b} = \begin{pmatrix} 0 & 0 & 0 & 0 \\ 0 & 1 & -1 & 0 \\ 0 & -1 & 1 & 0 \\ 0 & 0 & 0 & 0 \end{pmatrix}, \tag{C.1}$$

with $\beta = 2\cosh\eta = 2$.

The staggered transfer matrices are defined similarly (5.13),

$$\mathbf{T}^{\text{iso}}(0, \alpha) = (u + i)^{L/2}(u - \alpha + i)^{L/2}\text{Tr}_a\left[\prod_{m=1}^{L/2}\mathbf{R}_{a,2m-1}^{\text{iso}}(u)\mathbf{R}_{a,2m-1}^{\text{iso}}(u - \alpha)\right], \tag{C.2}$$

and the relation to the Floquet evolution operator (5.7) becomes

$$\mathbf{U}_{\text{F}}^{\text{iso}}(T) = \mathbf{U}_{\text{F}}^{\text{iso}}(\{0, \alpha\}) = \frac{1}{(i - \alpha)^L i^L}\left(\mathbf{T}^{\text{iso}}(0, \alpha)\right)^2 \mathbf{G}^2. \tag{C.3}$$

The relation between the inhomogeneity $\alpha$ and period $T$ has changed accordingly, i.e.

$$\frac{\alpha}{\mathrm{i} - \alpha} = \frac{\exp(2\mathrm{i}T) - 1}{2}, \text{ or } \alpha = \frac{\mathrm{i}(\exp(2\mathrm{i}T) - 1)}{\exp(2\mathrm{i}T) + 1}. \tag{C.4}$$

In this case, we have

$$\mathbf{U}_{\mathrm{F}}^{\mathrm{iso}}(T) = \prod_{m=1}^{L/2} \check{\mathbf{R}}_{2m-1,2m}^{\mathrm{iso}}\left(\frac{\mathrm{i}(1 - \exp(2\mathrm{i}T))}{\exp(2\mathrm{i}T) + 1}\right) \prod_{m=1}^{L/2} \check{\mathbf{R}}_{2m,2m+1}^{\mathrm{iso}}\left(\frac{\mathrm{i}(1 - \exp(2\mathrm{i}T))}{\exp(2\mathrm{i}T) + 1}\right). \tag{C.5}$$

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
