# Peer review of "The Floquet Baxterisation"

_SciPost Physics_

## Round 1 · Referee Report · Anonymous (Referee 1) · 2022-10-28

Report
This manuscript presents the construction and the study of integrable lattice models describing 1d quantum systems governed by Floquet Hamiltonians. The time-dependent Hamiltonians considered are periodic and piecewise constant with respect to time. The associated lattice models correspond to quantum circuits. They are constructed from an elementary R-matrix satisfying the Yang-Baxter equation, on a lattice with staggered spectral parameters. This construction, and the commutation relation with the staggered transfer matrix, are carried out for a generic R-matrix, without assuming the regularity and difference properties. Then, the six-vertex model with a staggering of order two is studied in detail. It corresponds to a stroboscopic XXZ Hamiltonian, which is Hermitian in the easy-axis regime |Delta|>1, and non-Hermitian but PT-symmetric in the easy-plane regime |Delta|<1. The finite-size spectrum of the Floquet Hamiltonian is derived through the Bethe Ansatz Equation. In the easy-plane regime, a phase transition is observed between two subregimes, which correspond to different eigenvalues of the ground state under an anti-unitary operator.
Before giving my comments, I want to indicate that, not being an expert in quantum circuits, I am not able to deliver a judgement on the interest of this manuscript from the perspective of quantum circuits, nor in the bibliographic review on quantum circuits given in the manuscript.
The general construction given Section 4 is quite elementary from the point of view of integrable models. The previous section gives convincing arguments that this simple construction deserves attention. The specific example of the six-vertex model in Sections 5-6 also involves simple calculations (i.e. writing the Bethe Ansatz equations and solving them numerically for small system sizes, and examining the relation between the spectral parameter alpha and the period T), but it is treated with care and clarity.
I appreciate the fact that in seemingly unphysical regimes of the six vertex model, e.g. |Delta|>1 and pure imaginary spectral parameter alpha, are given a nice physical interpretation, which is the quantum circuit.
Here is a list of issues I have identified in the manuscript, which deserve to be corrected:
-
In various places in the manuscript (in particular, in the abstract and in Section 3), it is asserted that the staggered six-vertex model is related in the scaling limit to the "black-hole" sigma model CFT. This is based on the results of [42-43], namely that the scaling limit of the staggered six-vertex model shares part of its operator spectrum and operator density with the "black-hole" sigma model CFT. However, in [45-46], with a more careful of the spectrum, using the IM/ODE correspondence, it was shown that the six vertex model actually corresponds to a different CFT.
-
There are sign mismatches in (5.5) and (5.6).
-
References [42] and [48] are identical. I suggest to cite also this study of the spectrum of the staggered six-vertex model through NLIEs : C. Candu and Y. Ikhlef, Non-Linear Integral Equations for the SL(2,R)/U(1) black hole sigma model J. Phys. A: Math. Theor. 46, 415401 (2013)
Overall, the manuscript is well written, and presents some interesting original material. Therefore, I recommend its publication in SciPost Physics, provided the above points are addressed.

---

## Round 1 · Referee Report · Anonymous (Referee 2) · 2023-1-17

Strengths
- rigorous results
Weaknesses
- style of presentation
Report
In this paper, the authors introduce a wide class of integrable quantum circuits built from inhomogeneous monodromy matrices obeying the Yang-Baxter equation. After providing a general overview, the authors outline the construction in the case of periodic and open boundary conditions. Subsequently, they detail the algebraic constructions associated with the inhomogeneous 6-vertex model. Lastly, the authors discuss the spontaneous breaking of the anti-unitary symmetry of the Floquet Hamiltonian in the easy-plane regime of the model.
In my impression, the presentation is somewhat eclectic. The manuscript walks the reader through a diverse range of interrelated topics, including the set-theoretic Yang-Baxter equation, Temperley-Lieb algebras, and various types of solutions with different boundary conditions. Certain connections to CFTs are highlighted as well. If I am not mistaken, this part is largely a summary of the existing literature, including recently published results by the authors themselves. I wonder how much value added value can these chapters be to non-expert readers with only a little background on the topics. For example, the connection between the first nontrivial charge and the black-hole sigma model related to certain non-compact CFTs could certainly be explained in more detail, or otherwise (assuming it is of marginal relevance) completely dropped. I also struggled with decoding the meaning of dynamical Floquet criticality and in what way is it relevant for the subsequent discussion. In my perspective, the most interesting, presumably new, result of this manuscript concerns the breaking of anti-unitary symmetry. This is however only discussed at the end of the paper in Section 5. My impression is that most of the content from Sections 2 to 4 does not go further than the standard application of the algebraic Bethe Ansatz with inhomogeneous transfer matrices, dating back to works by Faddeev and coworkers. It is however not easy to discern new results from the compilation of already-known results. I believe some improvements and clarifications will be valuable.
Other (technical) remarks to be addressed: - what does "Floquet-Baxterisation" signify? Is this a precise mathematical notion or some "poetic" name? I am only aware of the "Baxterisation" procedure, referring to promoting solutions to the constant Yang-Baxter equation with a complex spectral parameter. While Baxterisation is achieved at the level of R-matrices, Floquet pertains to inhomogeneities monodromy operators with staggered inhomogeneities. This type of construction with commuting staggered transfer matrices has indeed been around for quite some time, at least from the work by Destri and De Vega on light-cone discretizations. I see little reason to coin new names. - I failed to decipher the meaning of "when written in the representation of the XXZ Hamiltonian density, coincides exactly with the Hamiltonian of the lattice limit of the SL(2,R)/U (1) black hole sigma-model." Besides that, I cannot make sense of "a lattice limit" (of a QFT). I wonder whether these findings are the novel contribution of this work or perhaps quote some previous studies? What does it mean "it appears" (before eq. (3.4))? - I do not understand the statement about a novel dynamical Floquet criticality. I hope the authors can elaborate on it in the revised version. Another question related to the aforementioned CFT spectrum (cf. end of page 6): what does it mean for a discrete spectrum of Floquet Hamiltonain H_F to coincide with the spectrum of a non-compact CFT? - claiming that Ref. [17] computes multi-point correlators in (the integrable Trotterisation of) the XXX model seems quite a stretch. While in the quoted paper the authors cast the correlators in terms of a particular transfer matrix, they give no procedure for evaluating them at finite or asymptotically large times. - in eq. (3.13), the notation for the listed classes is not explained in the text - it is unclear what "the inhomogeneous transfer matrix with one additional spectral parameter" is referring to. Any object on a two-fold tensor space should possess two spectral parameters. - In Section 3.4, u_j represent operators. On the other hand, in Section 2 and in 4.1., the same notation is used to represent complex scalars pertaining to inhomogeneities of commuting transfer matrices. - I do not understand the statement regarding the connection between parameters u and T appearing just before eq. (5.6). Perhaps it was meant \alpha? - On page 18, the authors mention a conjecture from an earlier paper, reference [26], but it is unclear what precisely the conjecture is about. Moreover, after eq. (5.7), it is mentioned that the conjecture (again, without further details) has been only partially proven. Overall, I find Section 5.1. form rather obscure in its current form. - The comments made after eq. (5.19) are not particularly clear, or at least some additional references (about algebraic constructions of the Q operator) would probably be beneficial. - In Section 5.2, or preferably even earlier when first introducing the notion of "Floquet-Baxterisation", it would deserve to acknowledge some of the older works with staggered inhomogeneous row-to-row transfer matrices, particularly in the context of light-cone discretizations of integrable QFTs (sine-Gordon, sigma model etc.), e.g. by Destri and De Vega and Faddeev, Volkov and Reshetikhin. - At the end of page 24, I do not understand the third category (last bullet) of Bethe roots. Presumably, it is meant that roots organize into complex-conjugate pairs representing bound states, whereas the second category is single roots with a finite imaginary part?
Other comments: - a number of equations are missing a comma at the end - on pages 7 and 9: for consistency reasons, the spelling should be "Trotterised". - below eq. (4.2), "baxterised" should be capitalized - in eq. (4.16), there should be some space after the comma . in eq. (4.21), there is a missing comma after the ellipsis - in Fig. 9, there is a typo ("noted").
Requested changes
I would recommend a revision. I do not think it would be unreasonable to restructure the paper and only retain the necessary background information, or even attempt rewriting it from the ground up. I would suggest better stressing the new contributions (which should be briefly announced and discussed in the introductory sections) and focusing more on the key results and specifically on the symmetry breaking. Detours should be avoided as much as possible.

---

## Editorial Decision

unknown